JCB | Journal of Cell Biology

# A two-step search and run response to gradients shapes leukocyte navigation in vivo

Antonios Georgantzoglou[1], Hugo Poplimont[1], Hazel A. Walker[1], Tim Lämmermann[2], and Milka Sarris[1]

**Migrating cells must interpret chemical gradients to guide themselves within tissues. A long-held principle is that gradients guide cells via reorientation of leading-edge protrusions. However, recent evidence indicates that protrusions can be dispensable for locomotion in some contexts, raising questions about how cells interpret endogenous gradients in vivo and whether other mechanisms are involved. Using laser wound assays in zebrafish to elicit acute endogenous gradients and quantitative analyses, we demonstrate a two-stage process for leukocyte chemotaxis in vivo: first a "search" phase, with stimulation of actin networks at the leading edge, cell deceleration, and turning. This is followed by a "run" phase, with fast actin flows, cell acceleration, and persistence. When actin dynamics are perturbed, cells fail to resolve the gradient, suggesting that pure spatial sensing of the gradient is insufficient for navigation. Our data suggest that cell contractility and actin flows provide memory for temporal sensing, while expansion of the leading edge serves to enhance gradient sampling.**

## Introduction

Cell movement is a driving force in embryonic development, cancer cell dissemination, and immune responses. To reach their destination, migrating cells must accurately interpret chemical gradients in their environment. A general mechanistic principle is that gradients modulate actin polymerization at the cell front, such that leading edge protrusions are stimulated in the direction of the source ("local excitation"; Insall, 2010; Sarris and Sixt, 2015; Van Haastert and Devreotes, 2004; Xiong et al., 2010). Concurrently, protrusions are inhibited in other parts of the cell ("global inhibition") through long-range diffusion of chemical factors (Franca-Koh and Devreotes, 2004) or global mechanical tension (Diz-Muñoz et al., 2016; Houk et al., 2012), establishing cell polarity towards the gradient direction. In this framework, protrusive forces are the main output of chemoattractant gradient interpretation.

While the influence of chemical gradients on protrusions has been observed in many cells, it remains unclear whether this mechanism is sufficient to generate directed motion. Leading-edge expansion is driven by branching of actin networks through the actin nucleator Arp2/3 (Rotty et al., 2013). Yet, accumulating evidence indicates that loss of Arp2/3 does not compromise chemotaxis in a direct fashion. In cells moving in a mesenchymal-like fashion, like fibroblasts, Arp2/3 was found to be dispensable for locomotion per se, but important for co-ordinating optimal adhesions with the substrate (Wu et al., 2012; Case and Waterman, 2015). In amoeboid cells, such as

leukocytes, which can move in the absence of dedicated adhesion molecules under confinement (Lämmermann and Sixt, 2009; Paluch et al., 2016), Arp2/3 was found to be dispensable in 1D and 2D substrates (Leithner et al., 2016; Vargas et al., 2016; Wilson et al., 2013) but not in convoluted 3D matrices (Leithner et al., 2016). These observations suggest that protrusion expansion in amoeboid cells is important for exploration and discovery of optimal paths of least resistance rather as a driving force of movement. If protrusion extension is not a primary driver of cell locomotion, then how do chemical gradients redirect motion?

Actin flows have been proposed as a generic driving force for cell locomotion. Actin polymerization at the cell front is translated to protrusions to a certain extent, but cell contractility produces membrane tension that counteracts growing actin networks, resulting in retrograde flow of actin filaments (Callan-Jones and Voituriez, 2016; Case and Waterman, 2015). This generates a retrograde force that can be transduced to the substrate, through adhesion molecules or friction (Paluch et al., 2016), and propel the cell forward. Accordingly, the speed of retrograde actin flow is a determinant of cell speed (Maiuri et al., 2015). Interestingly, retrograde actin flows have also been linked with directionality of motion and chemoattractant signaling. First, actin flows enhance polarity by mediating transport of polarity factors (Goehring et al., 2011; Maiuri et al., 2015; Munro et al., 2004). Secondly, the coordination of actin flows in

[1]Department of Physiology, Development and Neuroscience, Downing Site, University of Cambridge, Cambridge, UK;   [2]Max Planck Institute of Immunobiology and Epigenetics, Freiburg, Germany.

Correspondence to Milka Sarris: ms543@cam.ac.uk.



different parts of the cell correlates with persistent motion (Yolland et al., 2019). Finally, the concentration of chemoattractant in the environment can influence the speed of actin flows (Hons et al., 2018). Together, these observations suggest that actin flows could play a key part in gradient responses. This idea has to some extent been explored in *Dictyostelium amoebae* that were artificially enlarged to create more demand for active transport mechanisms in maintaining polarity (Lange et al., 2016). However, it remains unclear how actin flows are altered by gradients in ordinary migratory cells within a multicellular organism.

How cells process the gradient in the first place is an additional matter of debate. Eukaryotic cells are capable of spatial sensing, whereby differences in attractant concentration are detected simultaneously across the cell, triggering internal signaling gradients that generate cytoskeletal polarity (Parent et al., 1998; Parent and Devreotes, 1999). This is often contrasted with bacterial chemotaxis, whereby cells resolve external gradients temporally, by making sequential comparisons while moving and prolonging their persistence while experiencing rising concentrations of attractant (Berg, 2008). However, eukaryotic cells also show evidence for temporal sensing, as they can stabilize polarity when experiencing rising concentrations of attractant (Albrecht and Petty, 1998; Skoge et al., 2014) and use random protrusions to make further temporal comparisons in attractant concentration (Andrew and Insall, 2007). The extent to which cells use spatial or temporal sensing to sense endogenous gradients in vivo is unclear. The mechanistic basis for how eukaryotic cells maintain memory of temporal comparisons is also unclear.

The disparity of observations in vitro regarding the role of protrusions, actin flows, and spatial sensing beg the question how cells ultimately interpret gradients in vivo. Here, we address this using zebrafish and mouse neutrophils as a model, acute chemotaxis assays, and quantitative analyses. By comparing the state of cells immediately before and after gradient exposure in vivo, we determined a two-stage process for gradient responses: First a "search" phase, during which wandering neutrophils abruptly decelerate and actin polymerization is stimulated at the cell front. Next, cells switch into a "run" phase, with fast actin flows, high speed, and directional persistence. When actin dynamics are perturbed, cells fail to resolve the gradient, indicating that pure spatial sensing is insufficient for navigation. Surprisingly, we found that contractile actin structures are more fundamental than protrusive forces for gradient interpretation (using submaximal inhibition for Arp2/3 and myosin so as to not abolish locomotion in vivo). Decreasing protrusive structures reduced the area of sampling and the accuracy of turns but did not affect search or run duration. By contrast, decreasing contractility improved the ability of cells to form oriented protrusions, yet the cells failed to stabilize polarity and kept searching unproductively. We propose that contractility provides memory for temporal sensing of gradients, at least in part by enabling enhancement of actin flows and stabilization of polarity upon experiencing a rise in attractant.

## Results

### Zebrafish and mammalian neutrophils respond to gradients via small turns and an increase in directional speed and persistence

To identify conserved effects of gradients on leukocyte motion patterns in vivo, we exploited a two-photon laser wound (LW) assay in zebrafish and mouse tissue. This provides acute activation of endogenous chemical gradients in a tissue and thus enables a direct comparison of cell behavior before and after gradient exposure. We first examined neutrophil gradient responses in zebrafish. We performed laser wounding and imaging by two-photon microscopy in the ventral fin (VF) of 3 d post-fertilization (dpf) transgenic larvae that express a fluorescent probe in neutrophils (Fig. 1 A and Video 1). The VF is devoid of neutrophils but located adjacent to the caudal hematopoietic tissue (CHT), a site of neutrophil residence and accumulation. We used a procedure that allowed us to collect data on neutrophil motion both before and after wounding in the same tissue (Fig. 1 A). Initially, we forced neutrophil exit from the CHT by adding the chemoattractant Leukotriene B4 (LTB4) in the medium (Coombs et al., 2019). This stimulates mobilization of neutrophils from the CHT into the VF, where the cells remain highly motile but responsive to secondary gradients by subsequent wounding (Coombs et al., 2019). After recording neutrophil motion for ∼30 min, we performed an acute LW. This led to rapid redirection of cell motion towards the wound focus (Fig. 1 B and Video 1).

To diagnose the effects of gradients on neutrophil behavior, we performed trajectory analyses. Previous studies indicated that cells moving in exogenous, constitutive gradients have biased speed in the direction of the source (Sarris et al., 2012). To explore whether acute endogenous gradients bias speed according to orientation, we plotted instantaneous speed against the cosine of the approach angle θ; this angle represents the angle between the motion vector and the vector that connects the initial timepoint of the motion vector with the closest point of the wound perimeter (WP; Fig. 1 C). Cosine of angle θ closer to 1 (θ close to 0°) means cell orientation towards the wound, while cosine of angle θ closer to –1 (θ close to 180°) means cell orientation away from the wound. We found that neutrophils increased speed towards the source and decreased speed in the opposite direction, in contrast to their movement prior to gradient exposure, where speed fluctuated independently of motion direction (Fig. 1 D). To quantify global changes in orientation, we plotted the cosine of the approach angle θ across different distances from the closest point of the WP. This revealed a marked increase in orientation bias towards the wound (higher average cosine of θ), as compared to the orientation of the same cells before wound, across the whole of the distance range measured (Fig. 1 E). Cells also showed some directional bias to this area in distances beyond 100 μm. Since this pattern was not specific to the post-wound condition, we reason it could reflect directional biases due to tissue geometry or other tissue gradients.

Quantification of average orientation of the cells does not distinguish whether neutrophils are performing oriented turns or changing their persistence. To distinguish these effects, we introduced another metric, angle δ, which represents the change

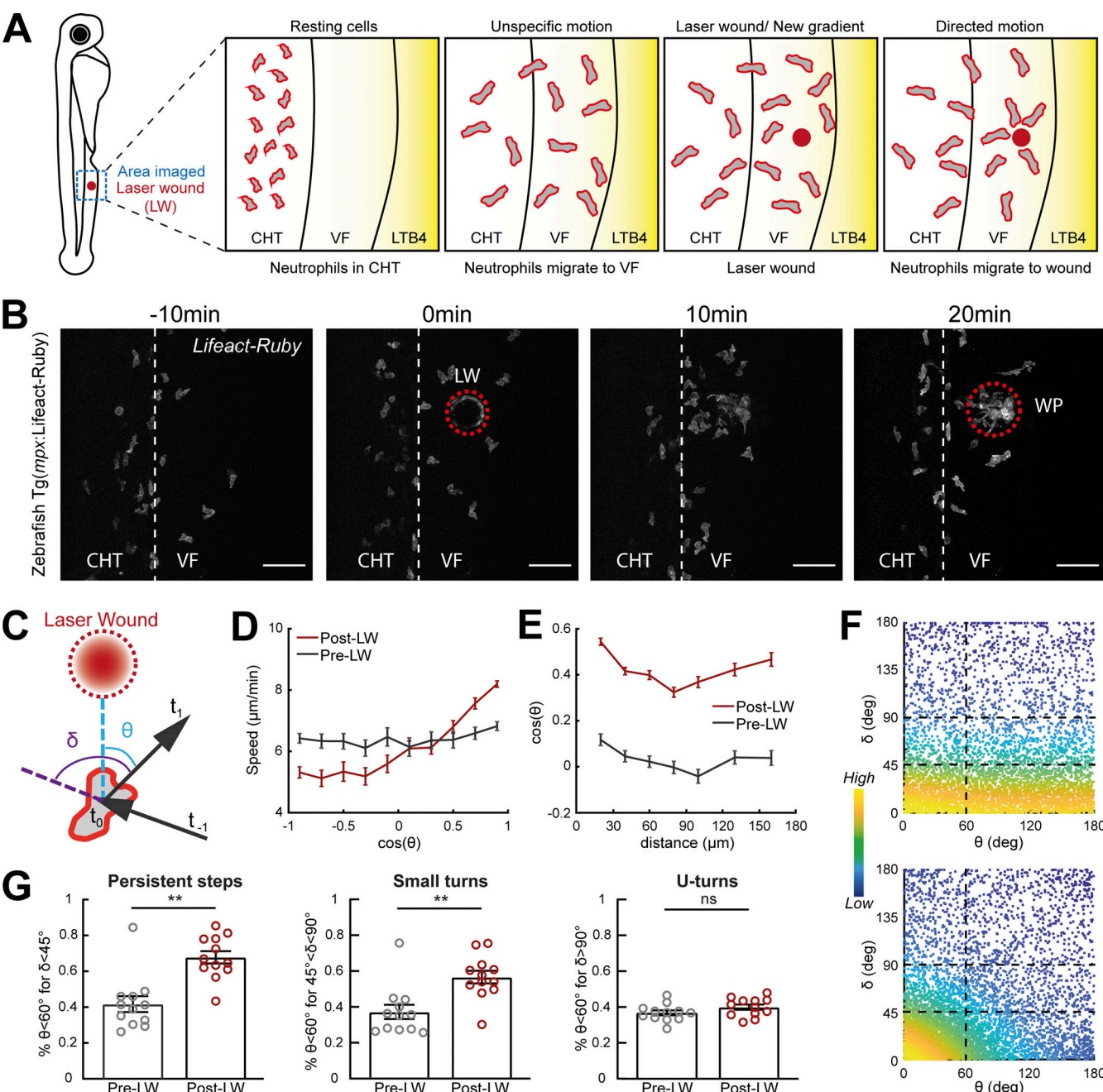

Figure 1. **Gradients in vivo cause a directional bias on speed and persistence and small reorientations towards the source.** Analysis of neutrophil migration responses to gradients in zebrafish. **(A)** Scheme showing two-photon LW chemotaxis assay in zebrafish. Resting neutrophils at the CHT are stimulated to exit the VF after LTB4 addition. Laser wounding in the VF at 30 min after LTB4 addition redirects neutrophil motion. **(B)** Time-lapse sequence of two-photon confocal image projections, showing fluorescent neutrophils in 3 dpf *Tg(mpx:Lifeact-Ruby)* transgenic larva, migrating pre- and post-LW (red dashed line). LW occurs at 0 min. Scale bar = 50 µm. **(C)** Scheme for angles δ and θ. δ is the angle between the speed vector of three successive timepoints $(t_0–t_{-1}, t_1–t_0)$. θ is the angle between the speed vector $(t_1–t_0)$ and the vector connecting the neutrophil geometrical center to the closest point of the LW perimeter. **(D)** Cell speed in relation to the cosine of θ for pre- and post-LW. n = 343–1,283 cell steps per bin pre-wound, n = 236–3,168 cell steps per bin post-wound. **(E)** Cell cosine of θ in relation to the distance from the closest point of the WP. n = 517–863 cell steps per bin for pre-wound, n = 444–1,272 cell steps per bin for post-wound. **(F)** Scatter plot of cell δ in relation to θ, color-coded for the density of points, for pre- (top) and post-LW (bottom). Black dashed lines indicate the groups of data points taken for analysis for G. Data shown from 12 larvae. **(G)** Percentage of cell persistent steps (δ < 45°), small turns (45° < δ < 90°) and U- turns (δ > 90°), respectively, with θ < 60°. Wilcoxon matched-pairs signed rank test, **, P = 0.0024 (persistent steps), P = 0.0068 (small turns). **(D, E, and G)** Data from 12 larvae in each case.

in angle between successive steps in neutrophil motion (Fig. 1 C). Cosine of angle δ closer to 1 (δ close to 0°) means no change of angle, while cosine of angle δ closer to –1 (δ close to 180°) means a cell U-turn. We distinguished a range of δ: persistent steps (defined here as δ < 45°), followed by small turns (defined here as δ in region 45–90°) and large U-turns (defined here as δ in region 90–180°; Fig. 1, F and G). Prior to gradient exposure, θ were distributed in an isotropic fashion across the range of δ, indicating no orientation bias in any of these steps (Fig. 1 G). After gradient exposure, we observed a significant increase in persistent steps and small turns but no increase in U-turns (Fig. 1, F and G). This is consistent with the idea that polarity has a certain memory and pre-polarized cells are resistant to polarity reversal (Albrecht and Petty, 1998; Arrieumerlou and Meyer, 2005; Gerisch and Keller, 1981; Skoge et al., 2014).

To test the conservation of these gradient effects in mammalian neutrophils, we analyzed data of two-photon imaging and laser wounding in mouse ear skin tissue using intravital microscopy (Fig. S1 A). We injected CellTracker Red CMTPX dye-labeled neutrophils that were isolated from bone marrow of C57Bl6 mice into the ear dermis of C57Bl6-Albino mice and waited for 4 h before intravital imaging was started. Previous studies have shown that neutrophil dynamics in this adoptive transfer model are comparable to the wound dynamics of endogenous mouse neutrophils (Lämmermann et al., 2013). Once imaging started, the motion pattern of neutrophils was recorded for 30 min before and after laser wounding. We observed marked chemotaxis and accumulation of neutrophils at the wound (Fig. S1 B and Video 2). Consistent with observations in zebrafish, neutrophils increased their directional speed and the probability of persistent steps and small turns in the direction of the source (Fig. S1, C–F).

Thus, neutrophils respond to newly encountered gradients in vivo by small adjustments to their migration path and a marked change in speed and persistence in the source direction.

## Front enrichment of actin polymerization probes correlates with slow motion and turning

To relate gradient-induced motion patterns to cytoskeletal dynamics, we analyzed actin dynamics in transgenic *Tg(mpx:Lifeact-Ruby)* zebrafish larvae (Yoo et al., 2010), whose neutrophils express Lifeact, a 17-amino-acid peptide that binds actin microfilaments (Riedl et al., 2008). Using the same two-photon LW assay, we followed the migration of neutrophils before and after wounding. We generated algorithms to automatically define the front and rear portions of the cells, based on the direction of motion and the geometrical center of the cells. We then computed Lifeact polarity as the ratio of fluorescence intensity at the front versus rear of the cell. Using cross-correlation analysis, we found that slow motion strongly correlated with front enrichment of Lifeact and, conversely, fast motion correlated with rear enrichment of Lifeact (Fig. 2, A–D). This correlation was observed both before and after laser wounding, suggesting that rear Lifeact configuration is generally associated with fast movement. To relate actin dynamics to cell directionality, we performed similar correlation analyses of Lifeact polarity against the δ metric (angle in relation to previous step, whereby small angle and higher cosine value indicates persistent

movement). Interestingly, while before laser wounding, there was no clear correlation between directionality and Lifeact distribution; in post-wounding, we detected a correlation between front Lifeact enrichment with cell turning (and thus of rear Lifeact enrichment with persistent steps; Fig. 2 E). This suggested that front enrichment is associated with periods of exploration that occur both in the presence and absence of gradients, but culminate in turning only in the presence of gradients.

## Rear enrichment of actin polymerization probes reports phases of fast actin flows

Front Lifeact enrichment is indicative of actin polymerization in this area and expansion of the leading edge (Riedl et al., 2008). However, the significance of rear Lifeact enrichment during fast motion remained less clear, as accumulation of the probe could report advection of the probe by actin flows as opposed to new polymer formation at the rear (Yamashiro et al., 2019). Accordingly, the rear Lifeact enrichment we observed could have reflected accelerated actin flows, as opposed to new actin polymer at the rear.

To test the relationship of actin flows to neutrophil motion in vivo, we performed spinning-disk confocal microscopy imaging with transgenic *Tg(lyz:DsRed2)$^{nz50}$/Tg(actb1:myl12.1-eGFP)* (Behrndt et al., 2012; Hall et al., 2007) zebrafish larvae, expressing nonmuscle Myosin-II (*myl12.1*) as well as a neutrophil DsRed marker. Myosin-II–GFP binds actin microfilaments with high affinity and as such has been particularly useful in tracking actin flows in vitro (Maiuri et al., 2015). To introduce directional stimulus, we utilized a mechanical wound assay (tail-fin [TF] amputation; Poplimont et al., 2020; Renshaw et al., 2006), as two-photon laser wounding was unavailable in the spinning-disk confocal microscopy setup. We followed the migration of neutrophils towards the TF within 1–2 h after wounding using high temporal resolution (Fig. 2 F and Video 3). We were able to discern flows using this probe in zebrafish neutrophils (Video 3), which were not as clearly visible using other actin-binding probes (*Tg[mpx:GFP-UtrCH]* or T*g[mpx:Lifeact-Ruby]*; Yoo et al., 2010; data not shown). This might be due to differences in binding kinetics of the probes to actin filaments in relation to the speed of flows in these particular cells. To track retrograde actin flows, we used particle image velocimetry (Fig. 2 F and Video 3; Betz et al., 2009; Davis et al., 2015). To quantify speed of retrograde flow in the reference system of the moving cell, we subtracted the centroid of surface-segmented neutrophil of every timeframe from the centroid of the first timeframe (Video 3). Plotting the speed of actomyosin flow against the speed of the cell across time for individual cells suggested a close correlation between the two parameters (Fig. S2 A). Cross-correlation analysis across many cells revealed that the speed of actomyosin flows in relation to the cell was tightly correlated to the speed of motion (Fig. 2 G). Conversely, the speed of actin flows was inversely correlated with cell turning (large δ, small cosine δ; Fig. 2 H and Fig. S2 B). We confirmed that fast motion also correlated with rear Lifeact distribution in this wound assay as in the LW assay, using *Tg(mpx:Lifeact-Ruby)* zebrafish larvae (Fig. S2, C and D).

These findings are consistent with previous in vitro measurements linking speed, flows, and polarity (Maiuri et al., 2015; Ruprecht et al., 2015) but also indicated that rear Lifeact

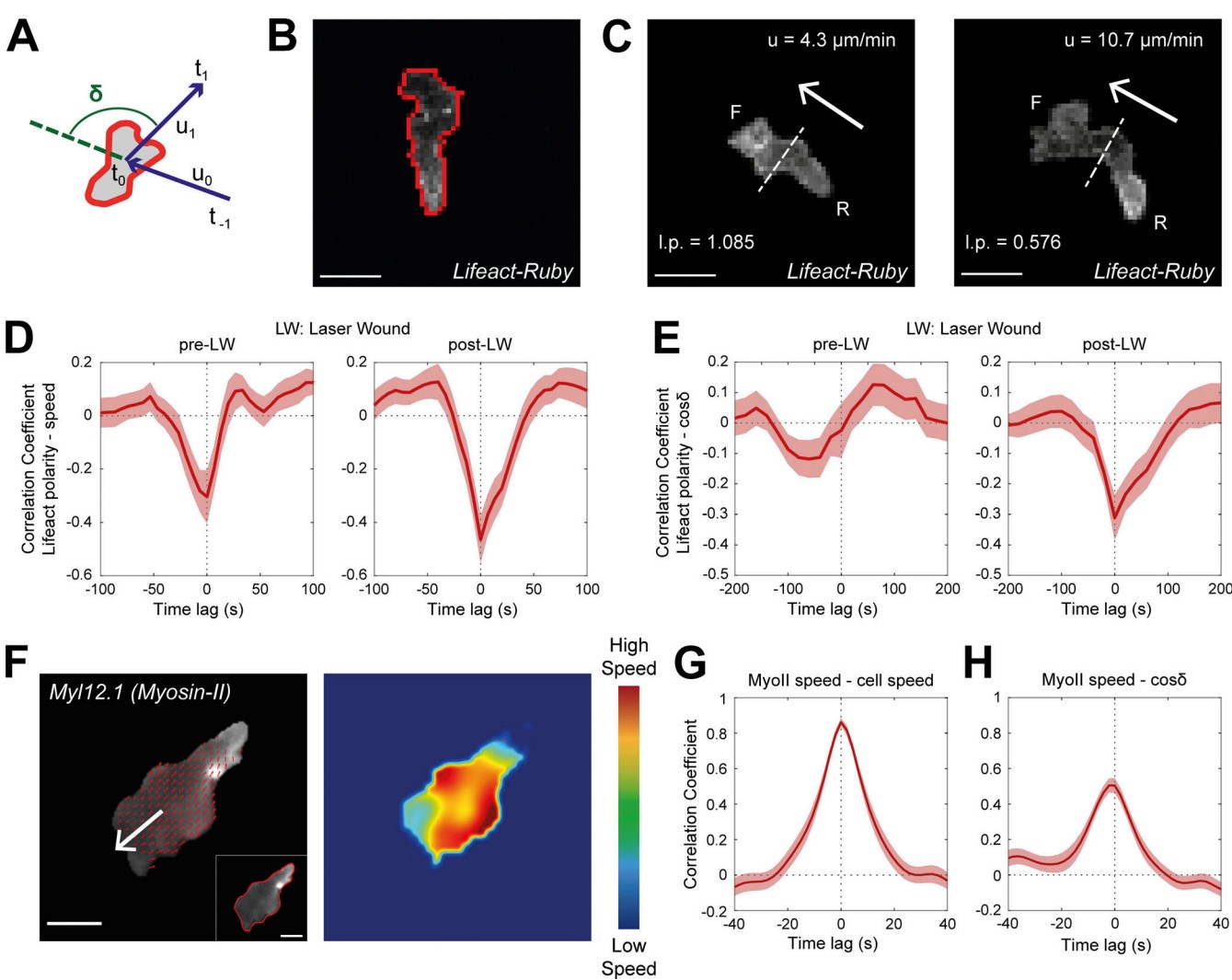

Figure 2. **Rear Lifeact distribution reports phases of fast motion and fast actin flows. (A)** Scheme of a neutrophil, indicating the definition of angle δ, between the velocity vectors (u) between three successive timepoints ($t_{-1}$, $t_0$, $t_1$). **(B)** Outline (red line) of an example of segmented neutrophil within a *Tg(mpx: Lifeact-Ruby)* transgenic larva. Scale bar = 10 µm. **(C)** Examples of neutrophil Lifeact distribution with an indication of cell speed (u) and Lifeact polarity (l.p.) at the same timepoint, for laser wounding. Arrows indicate the direction of motion. Dashed lines indicate the automated separation of the front (F) and rear (R) part of cell. Scale bar = 10 µm. **(D and E)** Temporal cross-correlation between Lifeact polarity and speed (D) or between Lifeact polarity and cosine of δ (E), pre- (left) and post-LW (right). Average from n = 21 migrating cells, from 9 larvae. Mean and SEM are shown. **(F)** Left: Representative neutrophil within a *Tg(actb1: myl12.1-eGFP)* transgenic larva migrating towards a mechanical wound; white arrow indicates the vector of speed; red arrows indicate the velocity vector fields of Myosin-II retrograde flow; inset (bottom right) indicates the segmented outline of the neutrophil. Scale bar = 5 µm. Right: Heatmap of the speed of the Myosin-II retrograde flow. Color bar indicates low- and high-speed values. **(G)** Temporal cross-correlation between Myosin-II retrograde flow speed and neutrophil speed. **(H)** Temporal cross-correlation between Myosin-II retrograde flow speed and neutrophil cosine of δ. **(G and H)** n = 25 cells, from 6 larvae. Mean and SEM are shown.

enrichment correlates with fast actin flows. Based on prior analyses (Yamashiro et al., 2019), this is most likely because of the advection of the probe, although we cannot exclude that some of this organization could also reflect new actin structures. This allowed us to use the rear Lifeact enrichment as a simplified proxy for detecting the state of fast actin flows, facilitating our subsequent analysis of the temporal sequence of gradient sensing events.

**Neutrophils respond to gradients first by leading-edge stimulation and subsequently by actin flow enhancement**
Thus far, we had identified gradient-induced motion patterns at the tissue-scale and associated subcellular actin dynamics. It

remained unclear how neutrophils respond to gradients precisely at the moment of gradient exposure and how relevant cytoskeletal dynamics evolve in time. To establish this, we profiled time-resolved changes in actin dynamics immediately before and after gradient exposure (Fig. 3). This analysis is difficult to conduct in vivo with direct measurement of actin flows, as it requires actin flow to be tracked in all cells for a consistent timescale before and after wounding (this is limited in these 3D environments because the imaging volume has to be small in order to image at high temporal resolution and cells exit the imaging volume at variable timescales). Another impediment was the absence of a two-photon laser wounding setup

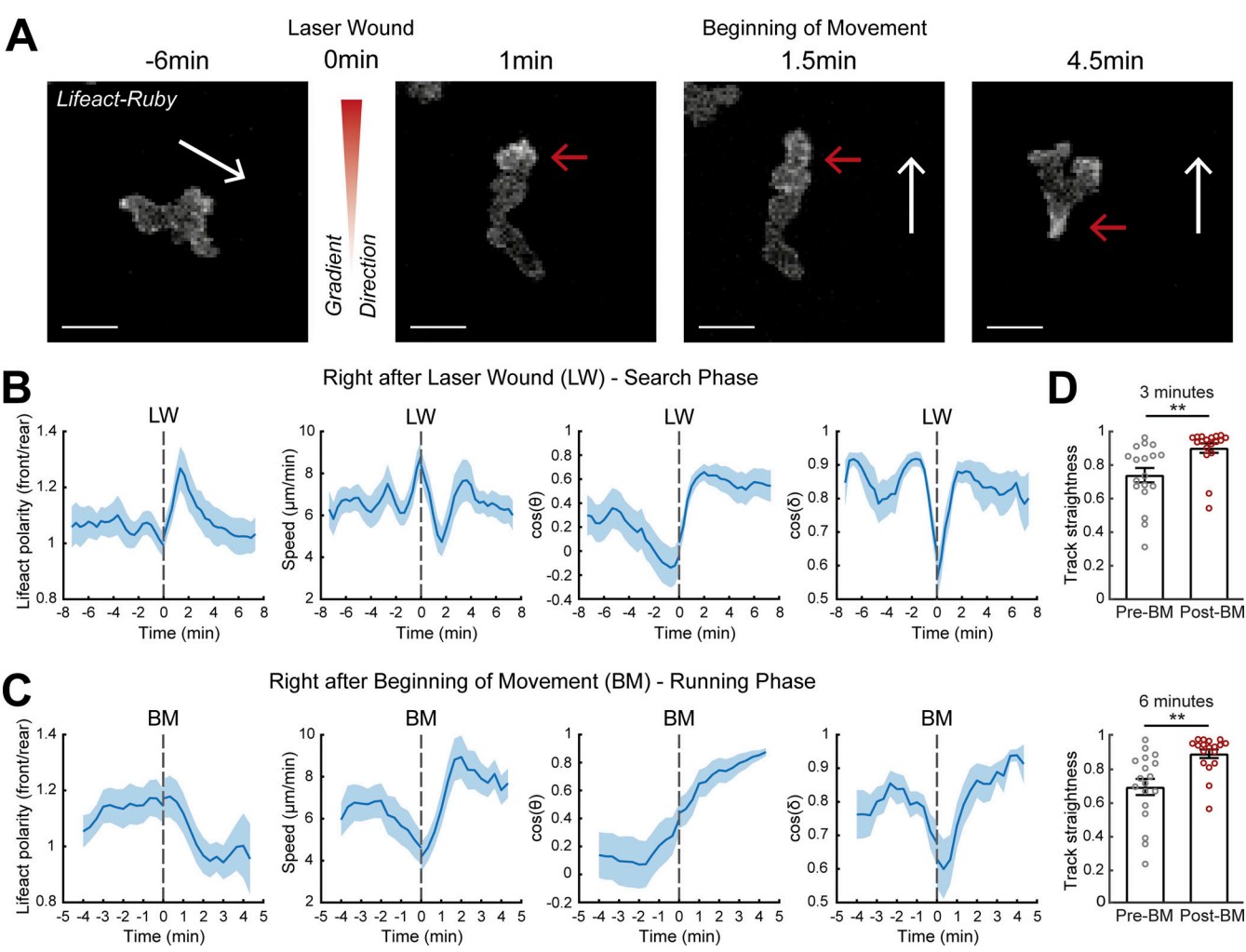

**Figure 3.  A two-stage response to newly encountered gradients in vivo. (A)** Time-lapse sequence of two-photon confocal image projections showing a neutrophil (white) in a *Tg(mpx:Lifeact-Ruby)* zebrafish larva, migrating pre- and post-LW. Cell movement started at 1.5-min after laser wounding for this example cell. White arrows indicate the speed vector. Red arrows indicate the side of cell with higher Lifeact abundance. Scale bar = 10 µm. **(B and C)** Neutrophil Lifeact polarity, speed, cosine of θ, and cosine of δ, in relation to time, respectively. Time sequence was synchronized based on time of the LW (B) or the time that each individual neutrophil began to move post-LW (C). **(B)** n = 21 cells, from 9 larvae. **(C)** n = 18 cells, from 9 larvae. **(D)** Neutrophil track straightness for motion completed within 3 (top) or 6 (bottom) min before and after beginning of neutrophil movement (BM). n = 18 cells, from 9 larvae. Mean and SEM are shown. Wilcoxon matched-pairs signed rank test, **, P = 0.0040 (3 min), P = 0.0019 (6 min).

coupled to a spinning-disk microscope. We therefore used the distribution of Lifeact as a simpler readout to infer on the status of actin flows, since our previous analysis showed how the two are linked. This revealed that when neutrophils first experience the gradient, within the first 2–4 min, they decelerated, concomitantly with a peak in front actin enrichment (Fig. 3, A and B; and Video 4 A). Towards the end of this phase, neutrophils completed a small turn in the direction of the source (change in δ and θ; Fig. 3 B). Subsequently, cells reinitiated movement in a nonsynchronized manner; such asynchrony may relate to the time needed to encounter signal and negotiate the direction of leading-edge protrusions. For the analysis, we synchronized cells to the timepoint of initiation of movement and we found that Lifeact was relocalized towards the rear of the cell concomitant with acceleration of cell motion and maintained persistent direction (stable δ; Fig. 3 C and Video 4 A). When quantifying the persistence of motion during a time window of 3

and 6 min before versus after the beginning of movement (first movement after LW), we found that cells increased their persistence of movement after gradient exposure (Fig. 3 D).

To corroborate these findings without exogenous factors for pre-stimulating motility (without LTB4), we performed live imaging of neutrophils pre- and post–laser wounding in the mesenchyme of the head (Fig. S2 E and Video 4 B), where a population of resident neutrophils displays constitutive, unspecific motility (Sarris et al., 2012; Yoo and Huttenlocher, 2011). We found a similar series of phases in the response of the cells, with a first stimulation of front Lifeact distribution that correlated with deceleration of motion and turning and subsequently a phase of fast and persistent motion (Fig. S2, F and G).

To corroborate these findings with an alternative actin probe, we used transient expression of a construct for Utrophin (*Tg [mpx:GFP-UtrCH]*) in *Tg(mpx:Lifeact-Ruby)* transgenic larvae (Yoo et al., 2010). While Lifeact stains all F-actin, UtrCH

(utrophin calponin homology domain) labels a subset of stable networks, which generally accumulate at the neutrophil tail (Yoo et al., 2010). We performed live imaging of these neutrophils following pre-stimulation of motility and laser wounding near the CHT as per our main assay (Fig. S3 and Video 5). We found that UtrCH showed a similar sequence of changes as compared to the Lifeact label (Fig. S3 A). Even though UtrCH polarity was biased towards the rear at all times (i.e., values of front/rear Lifeact polarity were below 1), the rear accumulation of this probe was reduced (higher front/rear Lifeact polarity) during the first phase (Fig. S3 B) and increased (lower front/rear polarity) during the second phase (Fig. S3 C). We reason that during the first slow phase, rearward actin flows are slow and therefore stable actin networks do not accumulate as fast in the tail. Thus, the findings with two different probes and chemotaxis assays were consistent.

Together these data indicate that we can distinguish two stages in the neutrophil gradient response: a first stage with active leading-edge actin dynamics and deceleration of motion, during which directional changes take place, and a second phase of fast actin flows and persistent and rapid motion.

## Gradient detection requires actin dynamics and active exploration

The gradient responses we observed could either result from purely spatial sensing of the gradient, i.e., simultaneous detection of the gradient at different parts of the cell independently from actin dynamics and motion, or from temporal sensing that requires sequential comparisons of the gradient in time through cell expansion or translocation (Insall, 2010; Parent et al., 1998; Parent and Devreotes, 1999; Sarris and Sixt, 2015; Servant et al., 2000). A key measure for this is to test whether signaling mediators downstream of chemoattractant receptors, such as the PI3K product PIP3, can polarize in the direction of the source in the absence of active exploration (Parent et al., 1998; Servant et al., 2000). To explore this in a live tissue context, we assessed the response of the signaling mediator PI3K after inhibition of actin polymerization with Latrunculin-B (LatB; Parent et al., 1998; Servant et al., 2000). We introduced LatB in the medium of transgenic Tg(mpx:PHAKT-EGFP) (Yoo et al., 2010) zebrafish larvae (whereby the PH domain of the kinase AKT is a probe for the PI3K product PIP3) after mechanical wounding and after the migration of neutrophils in this area had begun (Fig. S4 A and Video 6 A). Within 20 min, neutrophils were immobilized and many of them lost their polarized shape and became rounded (Fig. S4, A and B; and Video 6 A). Subsequently, we added LTB4 in the zebrafish medium to generate new exogenous chemical gradients in the VF to assess the signaling response of the immobilized cells (Fig. S4 A). We analyzed the distribution pattern of PIP3 before and after gradient exposure. We found that the orientation of PIP3, quantified as the ratio of fluorescence in the segment of the cell facing the source versus the opposite direction, was on average not biased towards the source direction (Fig. S4, B–D).

To complement these observations with endogenous gradients, we used LatB inhibition in neutrophils constitutively migrating in the head (Fig. 4 A and Video 6 B). Upon immobilization of the cells, we performed an LW, which leads to quick release of primary attractants, such as N-formyl-met-leuphe or ATP, immediately after wounding (Futosi et al., 2013; Kolaczkowska and Kubes, 2013; Poplimont et al., 2020). Analysis of PIP3 distribution before and after wounding again indicated that the orientation of PIP3 was not biased towards the source, consistent with the exogenous gradient assay (Fig. 4, B and C). To test whether PIP3 could be reoriented when actin dynamics were intact, we performed imaging of cells in the absence of LatB treatment (Fig. 4 D and Video 6 C). We observed two possible PIP3 responses in cells: (i) in cells with unipolar leading edge, PIP3 front to rear polarity could be enhanced after gradient exposure, and (ii) in cells with multipolar/expanded leading edge, PIP3 could be biased laterally towards the source (Fig. 4, E and F). Collectively, cells showed enhanced preference of PIP3 polarization towards the gradient source in these conditions.

Together, these results suggested that gradient detection relies on actin dynamics, indicating that pure spatial sensing is insufficient for gradient detection in vivo and that cells use actin dynamics to make sequential, temporal comparisons.

## Differential roles of protrusive and contractile actin structures in gradient sensing

As our analysis indicated that actin dynamics play a role in gradient sensing, we then investigated the role of protrusive and contractile forces in each aspect of the process. To address this, we visualized neutrophil motion and actin dynamics in transgenic Tg(mpx:Lifeact-Ruby) zebrafish larvae before and after gradient exposure in the presence of the Arp2/3 inhibitor CK666 and the Myosin-II inhibitor Blebbistatin, which inhibit protrusive and contractile actin networks respectively (Hetrick et al., 2013; Rauscher et al., 2018). We first exposed larvae to LTB4, to stimulate mobilization of neutrophils in the VF and unspecific motility in this tissue, followed by laser wounding to capture redirection of motion towards a new acute gradient. We found that at high enough concentrations both drugs caused a complete block of motility towards the source (Video 7), which for CK666 had also been previously reported in zebrafish neutrophils (Barros-Becker et al., 2017). We chose the maximal concentration that would not prohibit movement in both cases of drugs, to permit investigation of effects on gradient sensing. Whilst this could underestimate the effects of drugs, it would nevertheless permit the detection of specific contributions in gradient sensing versus more generic effects on locomotion.

Application of CK666 and Blebbistatin led to distinct cellular phenotypes in gradient responses. CK666-treated cells showed a decrease in surface area of the leading edge as compared with control cells (Fig. 5, A and B; and Video 8, A and B). This is consistent with the molecular role of Arp2/3 in 2D expansion of actin networks. By contrast, Blebbistatin-treated cells showed no significant change in surface area, although they displayed a more elongated shape due to an inability to retract the uropod (Video 8, A and C), consistent with previous observations (Yoo et al., 2010). Interestingly, Blebbistatin-treated cells showed a gain in ability to form an entirely new front towards the source from rear/lateral sides of the cells as opposed to modifying the leading edge (Fig. 5, A and C; and Video 8 D). This contrasts the

…

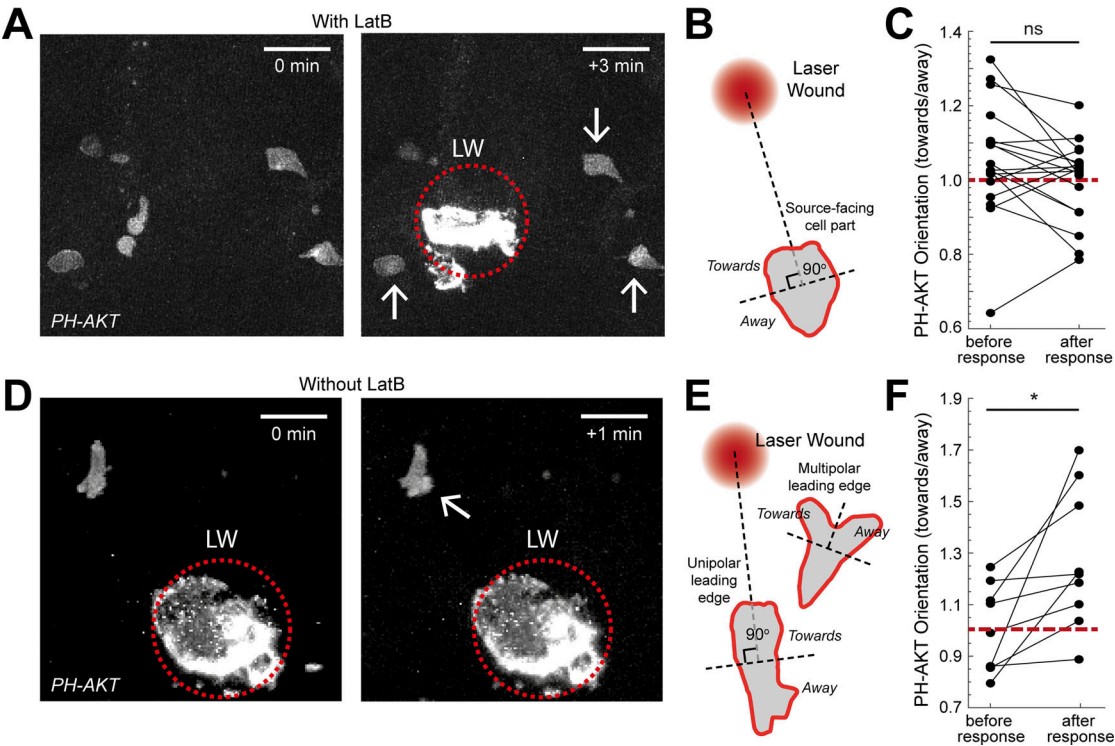

**Figure 4. Actin dynamics are required for gradient detection. (A and D)** Neutrophils in a *Tg(mpx:PHAKT-EGFP)* before and after their response to laser wounding (LW; red dotted circle), in the presence (A) or absence (D) of LatB. Arrow points to cells showing local increases in PIP3 intensity. Scale bar = 25 μm. **(B)** Scheme that shows segmentation of immobilized neutrophils in two parts, facing the source or the opposite direction. **(C)** Average neutrophil PIP3 orientation (towards/away) in LatB-treated neutrophils, before and after response to gradient. *n* = 18 cells from 6 larvae. Two-tailed paired *t* test. **(E)** Scheme that shows the segmentation of neutrophils, with either unipolar or multipolar leading edge into two parts, one facing the source or the opposite direction. **(F)** Average neutrophil PIP3 orientation before and after response to gradient (towards/away). *n* = 9 cells from 7 larvae. Two-tailed paired *t* test, *, P = 0.0176. **(C and F)** Red dashed line denotes orientation equal to 1.

pattern of untreated migrating cells here (Fig. 5, A and C; and Video 8, A and D) and of other pre-polarized cells studied elsewhere (Arrieumerlou and Meyer, 2005; Sarris and Sixt, 2015), whereby new extensions form preferentially at the leading edge. This indicates that contractility is important for memory of polarity, as its inhibition allows cells to more readily assemble an entirely new front. A possible mechanism that could account for these observations is that the loss of contractility at the rear weakens the actin cortex enabling protrusions to form more readily at the rear.

An additional mechanism responsible for the effect of Blebbistatin on the robustness of the polarity could be that contractility is important for enhancement of actin flows in the right direction (upon sensing rising attractant concentration). To quantify this indirectly, we compared the amount of time cells spent in search mode (with Lifeact enriched at the front—slow actin flow) versus in running mode (with Lifeact enriched at the back—fast actin flow) before and after gradient exposure (Fig. 5 D). We found no significant difference in control and CK666-treated cells in the amount of time in search and running mode. By contrast, Blebbistatin-treated cells significantly increased the amount of time in searching mode and decreased the amount of time in running mode, in the presence but not in the absence of the gradient. This suggests that in the absence of contractility, cells fail to enhance actin flows in the

right direction and get "stuck" into perpetual, unproductive searching.

We next investigated how these cellular phenotypes influence cell motion patterns. When assessing the probability of cells to perform oriented turns towards the source, we found that CK666-treated cells had a specific defect in turning that was not observed in Blebbistatin-treated cells (Fig. 5 E). When assessing cell speed, we found that CK666 and Blebbistatin both reduced motility regardless of the gradient presence (Fig. S5). However, Blebbistatin-treated cells showed a specific defect in biasing speed according to direction, as cells moved equally slow whether or not they were facing the source direction, whereas, like the control cells, CK666-treated cells showed relatively higher speed when moving towards versus away (Fig. S5 and Fig. 5 F). Finally, CK666 and Blebbistatin cells showed distinct effects on track straightness (Fig. 5 G). Whilst Blebbistatin-treated cells showed reduced track straightness in short (3 min) and long timescales (6 min) after beginning their movement after gradient exposure, CK666-treated cells showed a defect only in longer timescales (Fig. 5 G). We reason that the differential effects on track straightness relate to the distinct cellular phenotypes caused by the two treatments. CK666-treated cells make inaccurate turns the effect of which on cell trajectories would scale with longer timescales of observation, whereas Blebbistatin-treated cells have a defect in stabilizing/

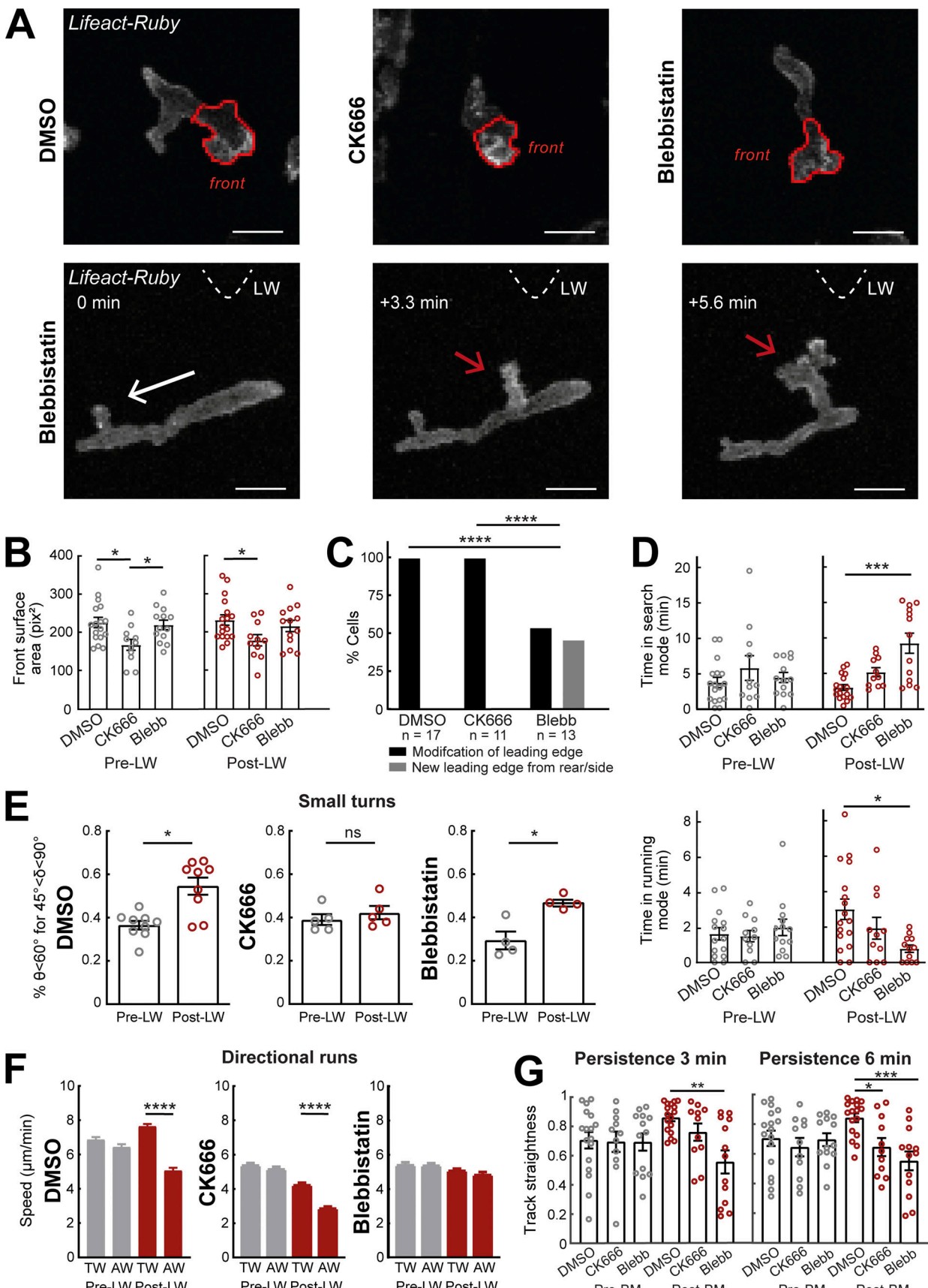

Figure 5. **Differential contributions of Arp2/3 and Myosin-II in gradient responses. (A)** Top row: Two-photon time-lapse image projections showing neutrophils in *Tg(mpx:Lifeact-Ruby)* zebrafish larvae treated with DMSO, CK666, and Blebbistatin, where the cell front has been segmented automatically (red

lines). Bottom row: Example neutrophil treated with Blebbistatin, with new leading edge forming at the cell lateral/rear towards the LW (red arrow). White arrow indicates the cell's direction of migration. Triangle indicates the direction of LW (not in the field of view). Scale bar = 10 µm. **(B)** Average surface area of the cell front part under DMSO, CK666, and Blebbistatin treatments. One-way ANOVA with Dunnett's multiple comparisons test, * P = 0.0138 (DMSO-CK666 pre-LW), P = 0.0437 (CK666-Blebbistatin pre-LW), P = 0.0379 (DMSO-CK666 post-LW). **(C)** Percentage of cells that generate a new leading edge from the rear or the side of their body, for the treatments of DMSO, CK666, and Blebbistatin, respectively. Chi-square test, ****, P < 0.0001. **(D)** Average time that cells spend in search (front Lifeact enrichment) and running (rear Lifeact enrichment) phase, for the treatments of DMSO, CK666 and Blebbistatin, respectively, for pre- (left) and post-LW (right). Kruskal-Wallis test with Dunn's multiple comparison test, ***, P = 0.0005 for time in search mode, *, P = 0.0124 for time in running mode. **(E)** Percentage of cells with persistent steps (θ < 60° for δ < 45°). n = 9 larvae for DMSO, n = 5 larvae for CK666, and n = 4 larvae for Blebbistatin. Wilcoxon matched-pairs signed rank test, *, P = 0.0195 for DMSO, P = 0.0216 for Blebbistatin. **(F)** Mean instantaneous speed for movement towards (TW) or away (AW) from the wound, for pre- and post-LW. n = 1,334–2,292 cell steps per condition for DMSO (9 larvae), n = 1,056–1,616 cell steps per condition for CK666 (5 larvae), and n = 818–1,826 cell steps per condition for Blebbistatin (4 larvae). Kruskal-Wallis test with Dunn's multiple comparison test, ****, P < 0.0001 for both DMSO and CK666. **(G)** Persistence of cells for 3 min (left) and 6 min (right) after neutrophil movement for DMSO, CK666, and Blebbistatin. One-way ANOVA with Dunnett's multiple comparisons test, **, P = 0.0004 (3 min, DMSO-Blebbistatin), *, P = 0.0154 (6 min, DMSO-CK666), ***, P = 0.0002 (6 min, DMSO-Blebbistatin). **(B–D and G)** Data from n = 17 cells from 8 larvae for DMSO, n = 11 cells from 5 larvae for CK666, and n = 13 cells from 4 larvae for Blebbistatin. Mean and SEM are shown.

memorizing polarity for a minimum amount of time after sensing the gradient, which would directly affect track straightness at any timescale.

Together, these findings suggest that leading-edge expansion is important for exploration and making the right choices during turning, whereas contractility is important for accelerating actin flows and thereby cell motion in the right direction and for stabilizing (keeping a memory of) polarity for a minimum timeframe after gradient sensing. Our data here also suggest that cells undergo qualitatively similar search phases before gradient exposure, with comparable surface area of leading edge (Fig. 5 B) and timescale of search (Fig. 5 D), but the result of these searches is different in the presence of gradients, with a bias in oriented turns (Fig. 2 D) and directional runs (Fig. 5 F).

## Discussion

Chemotaxis is fundamental for survival of unicellular and multicellular organisms. Current models are largely based on in vitro behaviors of isolated cells or unicellular organisms. How cells navigate in real tissue mazes within a live organism is poorly understood. This is important since in vivo cell behaviors and mechanisms are not always predicted from in vitro studies (Lämmermann et al., 2008). Here, we used high-resolution quantitative imaging and acute chemotaxis assays to describe the precise sequence of events when migrating cells encounter a new gradient in vivo. Surprisingly, we find that when a wandering cell experiences a new chemical gradient, the first response is to stop and explore (searching phase) rather than steer or accelerate. This is achieved through expansion of front actin networks, which promotes growth of leading edge and antagonizes cell motion. Cells then progress to the second stage of the gradient response, whereby contractile forces promote fast actin flows that stabilize motion in the direction of the source and provide a short-term memory of gradient direction (running phase; Fig. 6). We show that cells have limited capacity for spatially resolving gradients in vivo without active cytoskeletal dynamics. Our data indicate at least two ways by which actin dynamics support temporal gradient sensing in vivo: Protrusive actin structures mediate exploration and thus promote temporal sampling, while contractile forces are required for "memorizing temporal comparisons" by stabilizing polarity when moving up the gradient.

Using endogenously motile cells or pre-stimulation of motility, we revealed deceleration as unexpected first stage in the gradient response. Mechanistically, the loss of motility could be due to the transient uniform increase in attractant and receptor stimulation at the rear of the cell. We speculate that such output may also depend on the state of the cell upon encountering the gradient. If a cell is slow/resting, transient global receptor stimulation may readily translate into protrusions in all directions and no net change in motion. By contrast, in a fast-moving cell with pre-established actin flows, such receptor stimulation would in the first instance destabilize ongoing actin flows and thus motion speed. Our evidence indicates that these decelerations provide an opportunity for directional changes along the gradient. Previous studies have identified intrinsic phases of searching and running in random directions, for example the "run-and-tumble" behavior of germ cells (Reichman-Fried et al., 2004) and the "intermittent random walk" of dendritic cells (Chabaud et al., 2015). Here, we show that sensation of an external gradient can bypass intrinsic cell motion patterns and instruct a prompt, new search and an oriented run.

The degree to which eukaryotic cells use spatial sensing to interpret gradients within complex tissue environments in vivo has been unclear. While strong gradients could induce polarized signaling in immobilized *D. amoebae*, shallow gradients stimulated random protrusions that were selectively stabilized in the direction of the source (Insall, 2010; Parent et al., 1998; Parent and Devreotes, 1999). Our data indicate that neutrophils require actin dynamics to polarize their signaling machinery along the gradient. This suggests that, at least in some in vivo settings, gradients may be too weak to be resolved by pure spatial sensing, without exploration and memory. Though we have not directly measured cell responses to temporal gradients, our data with inhibition of contractility strongly suggest a role for memory in the response: These cells have increased ability to form oriented protrusions and revise their polarity when exposed to a new gradient but get "stuck" in perpetual searches and fail to stabilize motion. As contractility is important for actin flows and as actin flows are modulated by attractant concentration (Hons et al., 2018; Yolland et al., 2019), we propose that acceleration of actin flows when experiencing rise in attractant provides memory by selectively stabilizing polarity in the right direction. Given the general role of actin flows and contractile

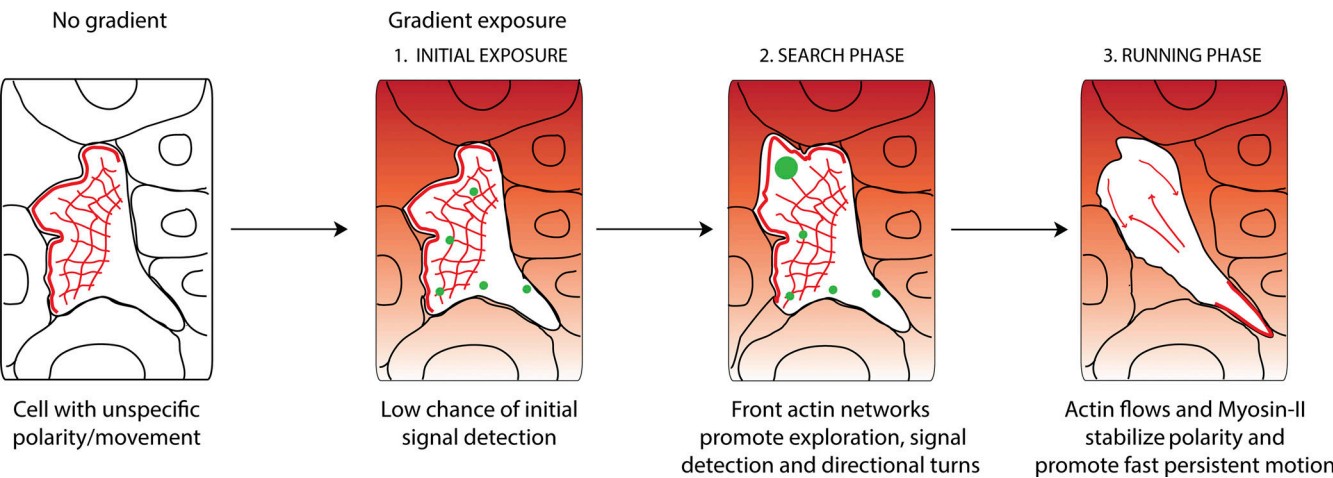

No gradient

1. INITIAL EXPOSURE

2. SEARCH PHASE

3. RUNNING PHASE

Gradient exposure

Cell with unspecific
polarity/movement

Low chance of initial
signal detection

Front actin networks
promote exploration, signal
detection and directional turns

Actin flows and Myosin-II
stabilize polarity and
promote fast persistent motion

Figure 6. **Neutrophils navigate interstitial gradients in vivo through a search and run strategy.** Model for neutrophil gradient sensing in vivo. Before encountering the gradient, neutrophils have a pre-established, unspecific polarity, and movement. Upon gradient exposure, neutrophils have a low chance of resolving gradient direction without active actin dynamics, therefore a search phase is required. During this phase, front actin network expansion via Arp2/3 promotes exploration and gradient sampling (illustrated by green spots), culminating in small turns towards the gradient source. Subsequently, cells switch to a running phase, whereby fast actin flows and Myosin-II contractility enforce fast and persistent motion towards the gradient.

forces in locomotion, we anticipate that our described role of contractility in gradient sensing is of relevance to other cell types. Recent work indicates that mouse macrophages can respond to gradients despite inhibition of Arp2/3 in vivo, suggesting that contractility-driven gradient processing is likely to be important in this context (Paterson and Lämmermann, 2022).

Our study reveals differential contributions of protrusive and contractile actin structures in the gradient sensing process. When using high enough doses of Arp2/3 and Myosin-II inhibitors, motility was perturbed, and we speculate that at such doses the protrusive and contractile forces are of insufficient level for actin flow generation. However, at submaximal doses that permitted movement, we could detect specific defects in gradient sensing. Consistent with previous models in *D. amoebae* (Andrew and Insall, 2007), we found that protrusions enhance exploration and enable correct turns, but this is only part of the gradient sensing process. Our data indicate that contractility allows cells to keep a short-term memory of gradient and accelerate and persist in a favorable direction. It can be postulated that this is more fundamental than leading-edge expansion. Firstly, reduction of leading-edge expansion by CK666 had an effect on trajectory shape that depended on the timescale of observation (over time cells accumulate wrong turning choices), whereas reduction of contractility affected persistence more directly, at any timescale. Secondly, limiting the leading-edge surface did not fundamentally disrupt the two phases of gradient response, whereas inhibition of contractility substantially increased the time spent in search mode versus running mode. Finally, previous studies in other leukocytes suggest the role of protrusions to be dispensable for chemotactic responses (Leithner et al., 2016; Paterson and Lämmermann, 2022; Vargas et al., 2016) and context-dependent (Leithner et al., 2016), as cells mostly benefit from an expanding leading edge in situations where there are many path options available (e.g., in complex 3D

tissue geometries rather than confined 1D tunnels). Together, these observations suggest that protrusive forces in gradient responses may be primarily important for generating rearward actin flows (which drive polarity, speed, and persistence) rather than for expanding protrusions (which drive exploration and accuracy). This would be consistent with the observation that front actin enrichment drives rearward flows and polarity in cells that move through blebs and thus do not require leading-edge expansion for directed motion (Olguin-Olguin et al., 2021).

Several questions remain in relation to gradient interpretation in vivo. Future work could be oriented towards exploring possible roles of tissue geometry on the search and run phases of gradient response, since the expansion of leading edge could become more important in certain tissue geometries. Another interesting question is how the transition from searching to running phase is achieved. One possibility is that once cells accumulate signal in one direction (signaling checkpoint model) this stimulates an active switch in contractility and an increase in actin flows. Another possibility is that fast actin flows naturally emerge when actin polymerization and local actin flows in different parts of the cell converge in the same direction (Yolland et al., 2019), while they might fail to emerge when initiated in antagonistic directions (emergent behavior model). Finally, it would be interesting to explore how the search and run pattern of gradient response is influenced by levels of receptors at the plasma membrane, since recent findings in neutrophils (Coombs et al., 2019; Kienle et al., 2021) and other cell types (Lau et al., 2020; Wong et al., 2020) indicate this to be an important factor in the behavior of cells in vivo.

Altogether, our study reveals how leukocytes employ leading-edge dynamics, contractility, and actin flows to interpret gradients within complex tissue settings. Given the fundamental role of chemotaxis in development, immunity, and cancer, our findings provide a paradigm of broad physiological relevance.

## Materials and methods

### General zebrafish procedures

The zebrafish lines used were: *Tg(mpx:Lifeact-Ruby)* (Yoo et al., 2010), *Tg(mpx:PHAKT-eGFP)* (Yoo et al., 2010), *Tg(actb1:myl12.1-eGFP)* (Behrndt et al., 2012), and *Tg(lyz:DsRed2)^{n250}* (Hall et al., 2007). Zebrafish were maintained in accordance with UK Home Office regulations, UK Animals (Scientific Procedures) Act 1986. Adult zebrafish were maintained under project licenses 70/8255 and P533F2314. Zebrafish were maintained according to Animal Research: Reporting of In Vivo Experiments guidelines. They were bred and maintained under standard conditions at (28.5 ± 0.5)°C on a 14-h light:10 h-dark cycle. Embryos were collected from natural spawning at 4–5 h after fertilization and thereafter kept in a temperature-controlled incubator at 28°C. Embryos were grown in E3 medium, bleached as described in the Zebrafish Book (Westerfield, 2007) and then kept in E3 medium supplemented with 0.3 µg/ml of methylene blue (Cat No. M9140-25G; Sigma-Aldrich) and 0.003% 1-phenyl-2-thiourea (Cat No. P7629-25G; Sigma-Aldrich) to prevent melanin synthesis. All embryos were used between 2.5 and 3.5 dpf, thus before the onset of independent feeding. For colocalization of Lifeact-Ruby and GFP-UtrCH, we injected the *Tol2(mpx:GFP-UtrCH)* (Yoo et al., 2010) plasmid in one-cell stage of *Tg(mpx:Lifeact-Ruby)* larvae. 1 nl solution was injected at each embryo; the solution contained 25 ng/µl DNA plasmid and 35 ng/µl Transposase mRNA. Where indicated, larvae were treated with: 1 µM LatB (Cat No. 428020-1MG; Merck), 30 nM LTB4 (Cat No. L0517-10UG; Sigma-Aldrich), 40–50 µM CK666 (Cat No. SML0006-5MG; Sigma-Aldrich), 150 µM Blebbistatin (Cat No. 203390-5MG; Merck), or DMSO (Cat No. D8418-100ML; Sigma-Aldrich).

### Two-photon LW and live imaging in zebrafish

For LWs, 3 dpf larvae were anesthetized with 0.04% MS-222 (Cat No. E10521-50G; Sigma-Aldrich) and mounted onto a glass-bottom plate in 2% low melting agarose (Invitrogen). Agarose-embedded embryos were covered with 2 ml E3 medium (supplemented with MS-222). Laser wounding was performed on a two-photon scanning miscroscope (LaVision Biotec TriM Scope II). A tunable ultrafast laser (Insight DeepSee, SpectraPhysics) was tuned to 900 nm and the laser power adjusted to ∼900 mW. Imspector software (Abberior Instruments) was used for image acquisition. A square region of interest (ROI) of 20–30 µm in width was defined in one focal plane, 10 µm below the larva surface, followed by single laser scan across the ROI at a pixel spacing of 240 nm and dwell time of 15 µs. Confocal stacks were acquired immediately after, using a 25× /1.05 numerical aperture (NA) water-dipping lens, at room temperature. GFP was imaged with 930 nm (for PIP3 analysis) and Ruby was imaged with a 1,040 nm (for Lifeact analysis).

### Mechanical wound and live imaging in zebrafish

For mechanical wounds, 3 dpf larvae were anesthetized with 0.04% MS-222 and their TF was amputated using a sterile surgical scalpel blade (Swann-Morton, 23). Larvae were mounted immediately after wound onto a glass-bottom plate in 2% low melting agarose (Invitrogen). Agarose-embedded embryos were covered with 2 ml E3 medium (supplemented with MS-222) and

imaged on a PerkinElmer UltraVIEW ERS spinning disk, Olympus IX81 inverted microscope, coupled to a Yokogawa CSU-X1 spinning-disk head and a Hamamatsu ORCA-Flash4.0 V2 sCMOS camera, controlled by Volocity software (PerkinElmer), equipped with a 30×/1.05 NA silicon oil (Olympus; for PIP3 and Lifeact analysis) or 60×/1.4 NA silicon oil objective (Olympus; for Myosin-II dynamics), using 488 nm laser for GFP excitation and 561 nm laser for DsRed excitation, at room temperature. Confocal stacks using a 2-µm z-spacing were acquired every 15 s for PIP3 analysis, every 10 s for Lifeact analysis, or every 0.5–2.5 s for Myosin-II analysis.

### General mouse procedures

C57Bl6 mice and C57Bl6-Albino (Tyr^{c-2J7c-2J}, JAX000058) mice were purchased from The Jackson Laboratory. Mice were maintained in specific pathogen–free conditions at an Association for Assessment and Accreditation of Laboratory Animal Care-accredited animal facility at the National Institute of Allergy and Infectious Diseases, National Institutes of Health, and were used under a study protocol from Dr. Ron Germain approved by National Institute of Allergy and Infectious Diseases Animal Care and Use Committee (National Institutes of Health). For intradermal injection experiments, mouse neutrophils were isolated from bone marrow of C57Bl6 mice using a three-layer Percoll gradient of 78, 69, and 52%. Neutrophils were washed three times with washing buffer (1× HBSS, 1% FBS, 2 mM EDTA). For fluorescent cell labeling, neutrophils were incubated for 15 min with 0.8 mM CellTracker Red CMTPX (Cat No. C34552; Invitrogen) in 1× HBSS supplemented with 0.0002% (wt/vol) pluronic F-127 (Thermo Fischer Scientific). Neutrophils were washed four times with washing buffer, before neutrophils (>2 × 10^6 cells) were taken up in 1× PBS at a volume of 15–30 µl. A volume of 5 µl neutrophil suspension was injected intradermally with an insulin syringe (31.5 GA needle; BD Biosciences) into the ventral side of the mouse ear pinnae. In some cases, they were coinjected with neutrophils from genetically modified mice that were not analyzed here, nor observed to affect behavior of wild-type neutrophils. Recipient mice were always on the C57Bl6-Albino background.

### Two-photon LW and live imaging in mice

Two-photon intravital imaging of ear pinnae of anaesthetized mice and laser-induced tissue injury was performed as previously described (Lämmermann et al., 2013). Analysis of mouse neutrophil dynamics was performed on experiments that were referred to, but data not shown in Lämmermann et al. (2013). 2–3 h after injection, mice were anesthetized and prepared for skin imaging and rested in the heated environmental chamber for 30 min before imaging started. Mice were anesthetized using isoflurane (Baxter; 2% for induction, 1–1.5% for maintenance, vaporized in an 80:20 mixture of oxygen and air), and placed in a lateral recumbent position on a custom imaging platform such that the ventral side of the ear pinna rested on a coverslip. A strip of Durapore tape was placed lightly over the ear pinna and affixed to the imaging platform to immobilize the tissue. Images were captured towards the anterior half of the ear pinna where hair follicles are sparse. Images were acquired using an inverted

LSM 510 NLO multiphoton microscope (Carl Zeiss Micro-imaging) enclosed in a custom-built environmental chamber that was maintained at 32°C using heated air, using Zeiss AIM 4.0 software. This system had been custom fitted with three external nondescanned photomultiplier tube detectors in the reflected light path. Images were acquired using a 25×/0.8 NA Plan-Apochromat objective (Carl Zeiss Imaging) with glycerol as immersion medium. Fluorescence excitation was provided by a Chameleon XR Ti:Sapphire laser (Coherent) tuned to 850 nm for excitation of CellTracker Red CMTPX dye-labeled neutrophils and second harmonic generation to orient in the connective tissue of the skin. For 4D data sets, 3D stacks were captured every 30 s. For focal tissue damage in the ear dermis, the Chameleon XR Ti:Sapphire laser (Coherent) was tuned to 850 nm and the laser intensity adjusted to 80 mW. At pixel dimensions of $0.14 \times 0.14$ μm, a circular ROI of 15–25 μm in diameter (approximately $1–2 \times 10^{-6}$ mm³ in volume) was defined in one focal plane, followed by laser scanning at a pixel dwell time of 0.8 μs for 35–50 iterations, depending on the tissue depth of the imaging field of view. Immediately after laser-induced tissue damage, imaging of the neutrophil response was started at typical voxel dimensions of $0.72 \times 0.72 \times 2$ μm³.

### Automated image analysis
#### Extraction of neutrophil trajectories
Analysis of neutrophil trajectories was performed in Imaris v8.2 (Bitplane AG) on 2D maximum intensity projections of the 4D time-lapse movies. Analyzed trajectories were extracted from the TF for mechanical wounds, VF (tracking data in the CHT were excluded) for two-photon ablations for Lifeact and GPF-UtrCH analysis, or area between eye and ear for two-photon ablations for Lifeact and PIP3 analysis. A track duration threshold of three timepoints was defined to exclude short-lived tracks. Manual track corrections were also applied where needed. Instantaneous neutrophil coordinates over time ($x$, $y$, $z$, $t$) were exported into Microsoft Excel 2016 spreadsheets files (Microsoft Corporation). The latter were imported into MATLAB R2018b (The MathWorks, Inc.) for processing and extraction of results.

#### Definition of mechanical and LW perimeter
For LWs, the perimeter of the wound was manually defined in MATLAB as a set of points surrounding the largest autofluorescent area around the wound. For mechanical VF wounds, the wound was manually defined in MATLAB as a set of points lining the edge of the amputated TF.

#### Neutrophil segmentation
For calculation of mean front and rear actin intensity and Myosin-II flow maps, neutrophil segmentation was achieved with active contours using the built-in MATLAB function for the Chan–Vese method (Chan and Vese, 2001) and custom-written MATLAB scripts (Coombs et al., 2019). When *Tg(actb1:myl1.2-eGFP)* fish were outcrossed with *Tg(lyz:DsRed2)*[nz50] fish, segmentation was done first on DsRed neutrophils using active contours, then the generated binary mask was applied on the Myosin-II neutrophils. When *Tg(mpx:Lifeact-Ruby)* were injected with GFP-UtrCH, segmentation was done first on the Lifeact-Ruby neutrophils using active contours, then the generated binary mask was applied on the GFP-UtrCH neutrophils. Segmentation of immobilized neutrophils in *Tg(mpx:PHAKT-EGFP)* larvae was achieved with active contours, whereas mobile neutrophils where segmented manually.

#### Definition of Lifeact or Utrophin polarity
For calculation of Lifeact or Utrophin polarity, the separation of a neutrophil into front and rear part was achieved using MATLAB with custom-written scripts. The front and rear parts of neutrophils were defined based on the vector of speed between two successive neutrophil centroids, as calculated from segmentation. A vertical to the speed vector line was defined to separate the neutrophil into two parts. The center of the line was the centroid of the segmented neutrophil. The front part was formed by grouping pixels with distance from the line higher than zero, while the rear part was formed by grouping pixels with distance from the line below zero. Time plots of Lifeact or Utrophin polarity were performed using the MATLAB function *boundedline* (Kearney, 2022).

#### Quantification of PH-AKT dynamics
For calculating the PH-AKT mean intensities, we used custom-written MATLAB scripts, including MATLAB's built-in function *nanmean*. Cell orientation was calculated as the ratio of mean intensities of "towards" and "away" neutrophil parts. For separation of individual immobilized neutrophils into towards and away segments, a line was defined to connect the geometrical neutrophil centroid with the nearest point of the mechanical TF wound or the center of the LW. A second line, perpendicular to the first line, was defined as passing from the geometrical centroid. The towards part was formed by grouping pixels with distance from the line higher than zero, while the away part was formed by grouping pixels with distance from the line below zero. Analysis was based on timepoints just before LTB4 addition (–0.25 min) or laser wounding (–0.5 min) and a timepoint within the first 3 min after LTB4 addition/LW, at which maximal visible change in PH-AKT intensity was observed. For separation of individual mobile neutrophils, manual segmentation was performed. For cells with unipolar leading edge, the towards and away parts were defined as for the immobilized cells. For cells with multipolar leading edge, the towards part was defined as the part of the leading edge facing the LW and the away part was defined as the part of the leading edge facing away from the LW. Analysis was based on custom timepoints between timepoints that cells showed change of direction towards the LW.

#### Quantification of directional speed, track straightness, angles θ and δ, and actomyosin flows
The calculation of speed versus orientation (directional speed) and of track straightness was achieved using custom-written MATLAB scripts (Sarris et al., 2012). The instantaneous speed was calculated using the distance that a neutrophil traveled between two successive timepoints (Coombs et al., 2019). For laser-wounding experiments, the timepoint of beginning of neutrophil movement was selected individually for each

neutrophil, based on the timepoint that each neutrophil began the movement that led to maximum speed of the track towards the wound. Track straightness was calculated as the ratio of net linear distance traveled by a neutrophil between the first and last timepoint of a defined time window to the total distance traveled in that time window. Neutrophil approach angle θ and turning angle δ were calculated using custom-written MATLAB. Instantaneous neutrophil θ was calculated as the angle between the vector of instantaneous speed and the vector that connects the first centroid and the nearest point of the WP (Coombs et al., 2019). Instantaneous neutrophil δ was calculated using three timepoints and was defined as the angle between the vector of instantaneous speed between first-second timeframes and the vector of instantaneous speed between first-third timeframes. Scatter plots of angle δ in relation to angle θ were performed using the MATLAB function *dscatter* (Henson, 2022). Time plots of speed, θ and δ, and correlation plots were performed using the MATLAB function *boundedline* (Kearney, 2022).

For calculation of retrograde flow maps, migrating neutrophils were transformed into stationary ones, by subtracting the coordinates of their centroid in each timeframe from the coordinates of their centroid in the first timeframe. The calculation of the Myosin-II retrograde flow maps was done using particle image velocimetry (Davis et al., 2015). According to this method, the cell cytoplasm was separated into small parts, and we tracked cellular parts with similar intensity between two successive timeframes, in a defined neighborhood. We chose the Myosin-II parts with an angle between their speed vector and the vector of whole neutrophil speed above 145° and below 225° (in the range 0–360°). Correlation coefficients were calculated by custom-written MATLAB scrips, using MATLAB's built-in cross-covariance function *xcov* (Mueller et al., 2017).

### Statistics
All error bars indicate SEM. All P values were calculated with two-tailed statistical tests and 95% confidence intervals. Two-tailed paired $t$ test and Wilcoxon matched-pairs signed rank test were performed for pairwise comparisons, and ordinary one-way ANOVA with Dunnett's multiple comparisons test was performed after distribution was tested for normality with the Shapiro-Wilk test; otherwise, Kruskal-Wallis test with Dunn's multiple comparison test was used instead of ANOVA. Statistical tests were performed in Prism v9.3.1 (GraphPad Software, Inc.). The statistical test and the $n$ number are indicated in the figure legends. The error bars show SEM either across individual embryos (i.e., analysis of population neutrophil persistence turning) or individual neutrophils pooled from different embryos (i.e., speed versus orientation and orientation versus distance from wound, which are step-based analyses, or speed versus time, Lifeact polarity versus time, orientation versus time, turning versus time, correlation coefficient, track straightness, cell orientation, front surface area, time in search or running mode, which are cell-based analyses).

### Online supplemental material
Fig. S1 (in support of Fig. 1) reports that neutrophil migration responses to gradients in mice are similar to those in zebrafish.

Fig. S2 (in support of Figs. 2 and 3) shows the two-stage process in gradient sensing that is observed in endogenously motile neutrophils. Fig. S3 (in support of Fig. 3) reports the distribution of Utrophin marker during neutrophil response to gradient. Fig. S4 (in support of Fig. 4) shows that actin dynamics are required for gradient detection in neutrophils. Fig. S5 (in support of Fig. 5) reports that Myosin-II inhibition impairs directional bias on speed. Video 1 (in support of Figs. 1 and 2) shows the neutrophil migration pre- and post-LW in a zebrafish larva. Video 2 (in support of Fig. 1) shows the neutrophil migration pre- and post-LW in a mouse. Video 3 (in support of Fig. 2) shows the retrograde Myosin-II flows in zebrafish neutrophils when migrating towards a wound. Video 4 (in support of Figs. 3 and S2) shows the response of an individual neutrophil to LW. Video 5 (in support of Fig. S3) shows the Lifeact-Ruby and GFP-UtrCH distribution of an individual migrating neutrophil. Video 6 (in support of Figs. 4 and S4) shows the PH-AKT distribution in immobilized neutrophils before and after gradient exposure. Video 7 (in support of Figs. 5 and S5) shows that high dose of CK666 and Blebbistatin disrupts locomotion in neutrophils. Video 8 (in support of Figs. 5 and S5) shows the effects of inhibitors CK666 and Blebbistatin on leading edge expansion in neutrophils.

## Acknowledgments
We thank Rob White for comments on the manuscript; Kevin O'Holleran and Martin Lenz of the Cambridge Advanced Imaging Centre for their support and assistance in two-photon microscopy; Bill Harris, Christine Holt, and Ewa Paluch groups for access to spinning-disk confocal microscopy equipment; Ronald Germain for intravital microscopy equipment; fish facility staff for assistance with zebrafish husbandry; Andrei Luchici and Brian Stramer (King's College London, London, UK) for sharing algorithms for analysis of actomyosin flows; Anna Huttenlocher (University of Wisconsin Madison, Madison, WI) for the *Tg(mpx: Lifeact-Ruby)* and *Tg(mpx:PHAKT-EGFP)* zebrafish lines and the *Tol2(mpx:GFP-UtrCH)* construct; Carl-Philipp Heisenberg (Institute of Science and Technology Austria, Klosterneuburg, Austria) for the *Tg(actb1:myll2.1-eGFP)* line; Phil Crosier (University of Auckland, Auckland, New Zealand) for the *Tg(lyz:DsRed2)[nz50]* line.

M. Sarris, A. Georgantzoglou, and the research were supported by a Medical Research Council Career Development Award (MR/L019523/1); Wellcome Trust (204845/Z/16/Z); Isaac Newton Trust (12.21 [a]i and 19.23 [n]), a Physiological Society research grant, and a Leverhulme Trust grant (RPG-2021-226). H.A. Walker was supported by a Medical Research Council Doctoral Training Partnership program. H. Poplimont was supported by a Wellcome Trust PhD grant (105391/Z/14/Z). T. Lämmermann was funded by the Max Planck Society.

The authors declare no competing financial interests.

Author contributions: Conceptualization: M. Sarris. Methodology: M. Sarris. Investigation: A. Georgantzoglou, T. Lämmermann, H. Poplimont, H.A. Walker. Formal analysis: A. Georgantzoglou. Software: A. Georgantzoglou. Visualization: A. Georgantzoglou. Writing: M. Sarris. Writing—review & editing: M. Sarris, A. Georgantzoglou, T. Lämmermann. Funding

Acquisition: M. Sarris, T. Lämmermann. Supervision and Project Administration: M. Sarris.

Submitted: 31 March 2021

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

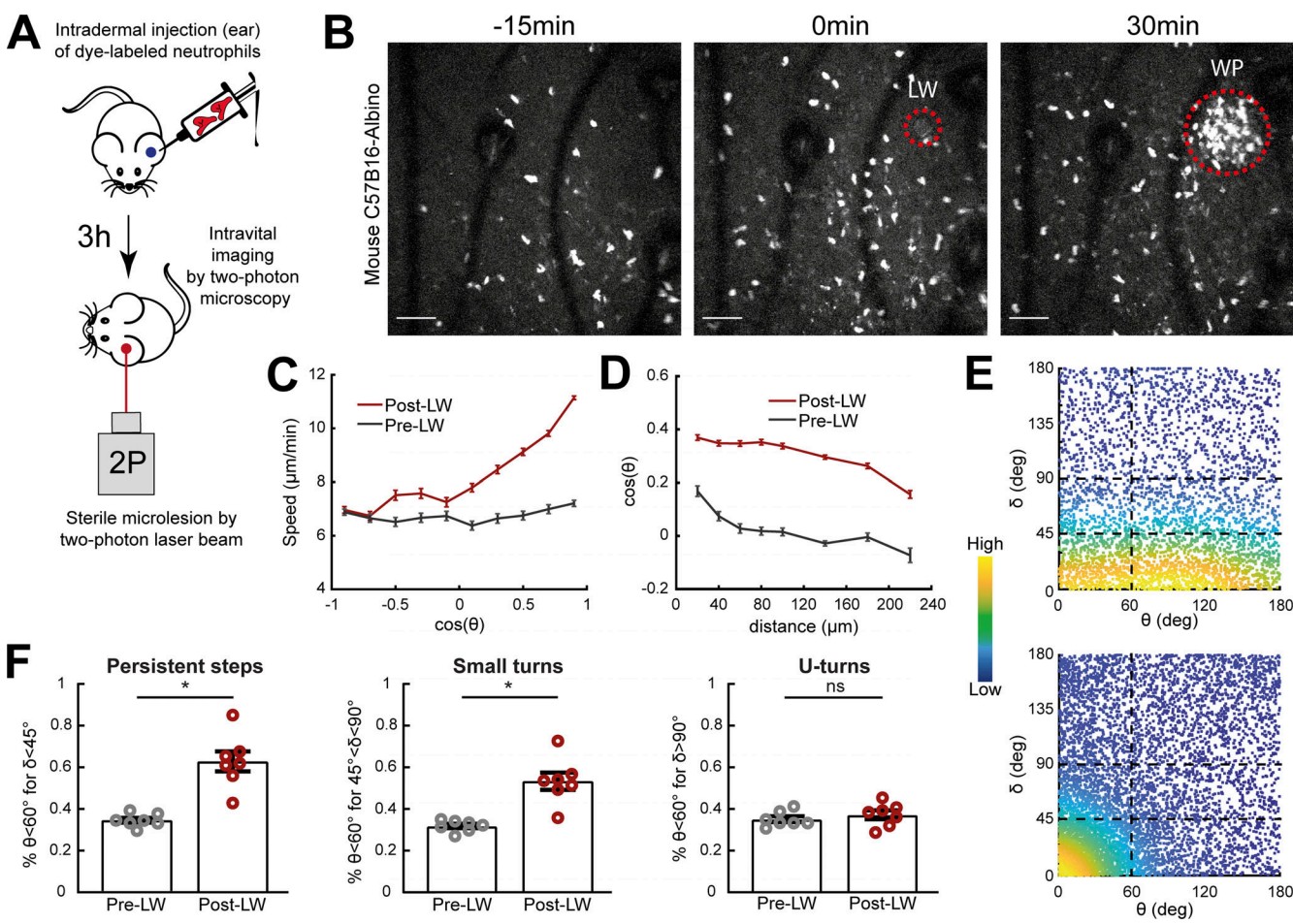

Figure S1. **Neutrophil migration responses to gradients in mice are similar to those in zebrafish.** In support of Fig. 1. **(A)** Scheme showing two-photon LW chemotaxis assay in mice. Neutrophils from bone marrow of C57Bl6 mice were isolated, fluorescently dye-labeled, and then injected into the ear dermis of C57Bl6-Albino mice. 3 h later, intravital imaging started by recording neutrophil dynamics before and after induction of a small tissue lesion in the skin dermis by a two-photon laser beam. **(B)** Time-lapse sequence of two-photon confocal image projections, showing CMTPX dye-labeled neutrophils (white) in a C57B16-Albino mouse, migrating pre- and post-LW (red dashed line). LW occurs at 0 min. Scale bar = 50 μm. **(C)** Cell speed in relation to the cosine of θ for pre- and post-LW. n = 629–4,879 cell steps per bin for pre-wound, n = 2,152–6,673 cell steps per bin for post-wound. **(D)** Cell cosine of θ in relation to the distance from the closest point of the WP. n = 1,145–3,156 cell steps per bin for pre-wound, n = 1,580–12,000 cell steps per bin for post-wound. **(E)** Scatter plot of cell δ in relation to θ, color-coded for the density of points, for pre- (top) and post-LW (bottom). Black dashed lines indicate the groups of data points taken for analysis for F. Plotted data from two representative mice. **(F)** Percentage of cell persistent steps (δ < 45°), small turns (45° < δ < 90°), and U-turns (δ > 90°), respectively, with θ < 60°. Wilcoxon matched-pairs signed rank test, *, P = 0.0156 (for both persistent steps and small teurns). **(C, D, and F)** Data from seven mice.

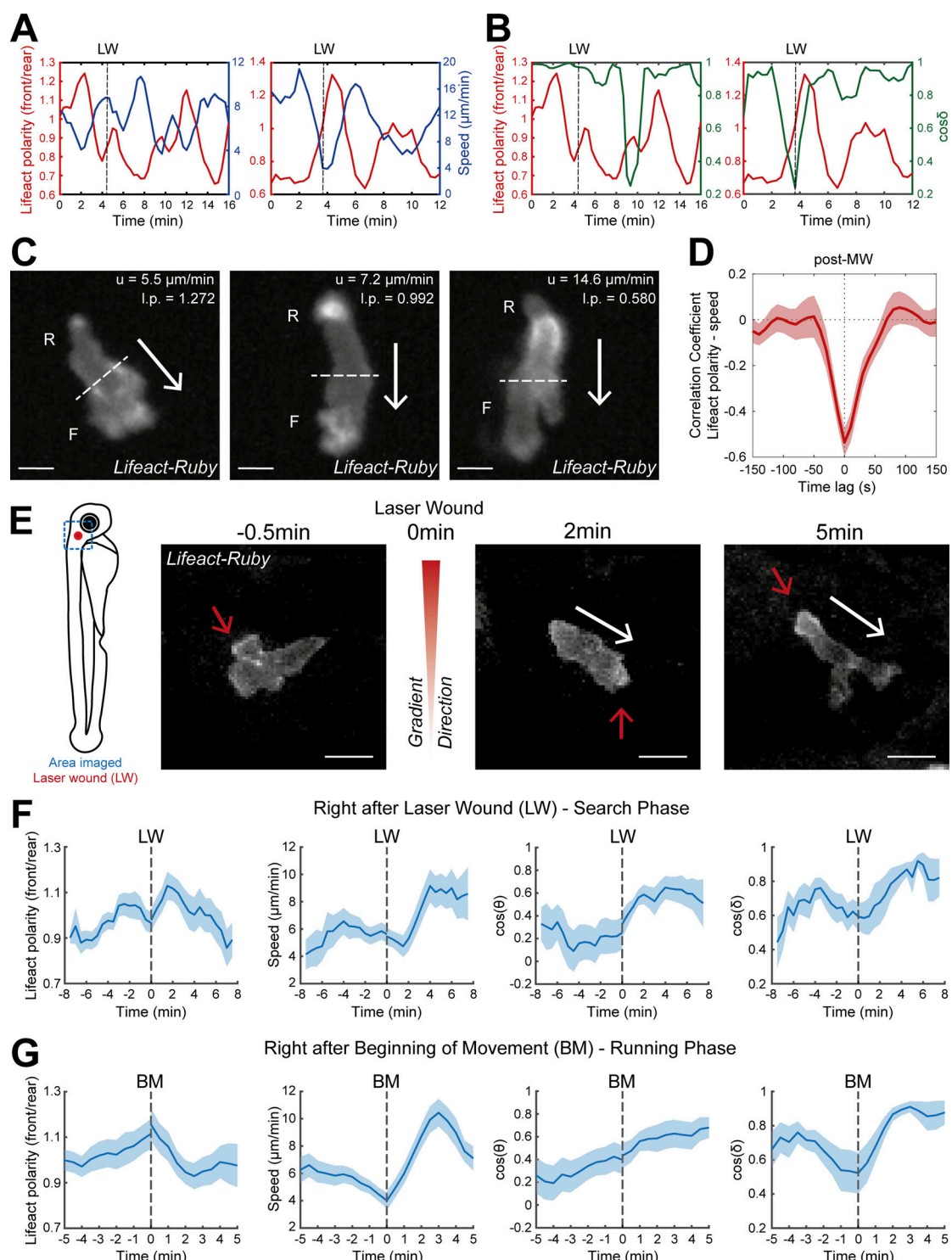

Figure S2. **A two-stage process in gradient sensing is observed in endogenously motile neutrophils.** In support of Figs. 2 and 3. **(A and B)** Example of time evolution of Lifeact polarity (red color) with neutrophil speed (blue color; A) and cosine δ (green color; B) from two individual neutrophils. LW: time of laser wounding. **(C)** Examples of neutrophil Lifeact distribution with an indication of cell speed (u) and Lifeact polarity (l.p.) at the same timepoint, for mechanical wounding. Arrows indicate the direction of motion. Dashed lines indicate the automated separation of the front (F) and rear (R) part of cell. Scale bar = 5 μm. **(D)** Temporal cross-correlation between Lifeact polarity and speed post–mechanical wounding. Average from n = 14 migrating cells, from 2 larvae. Mean and SEM are shown. **(E)** Scheme of zebrafish larva with indication of laser wounding area (red dot) and imaging area (blue dashed square). Time-lapse sequence of two-photon confocal image projections showing a neutrophil (white) in a *Tg(mpx:Lifeact-Ruby)* zebrafish larva, at 0.5 min pre–laser wounding and at 1 and 5 min post–laser wounding. White arrows indicate the speed vector. Red arrows indicate the side of cell with higher Lifeact abundance. Scale bar = 10 μm. **(F and G)** Neutrophil Lifeact polarity, speed, cosine of θ, and cosine of δ, in relation to time, respectively. Time sequence was synchronized based on time of the LW (F) or the time that each individual neutrophil beginning of movement post–laser wounding (G). **(D)** n = 21 cells, from 9 larvae. **(E)** n = 18 cells, from 9 larvae.

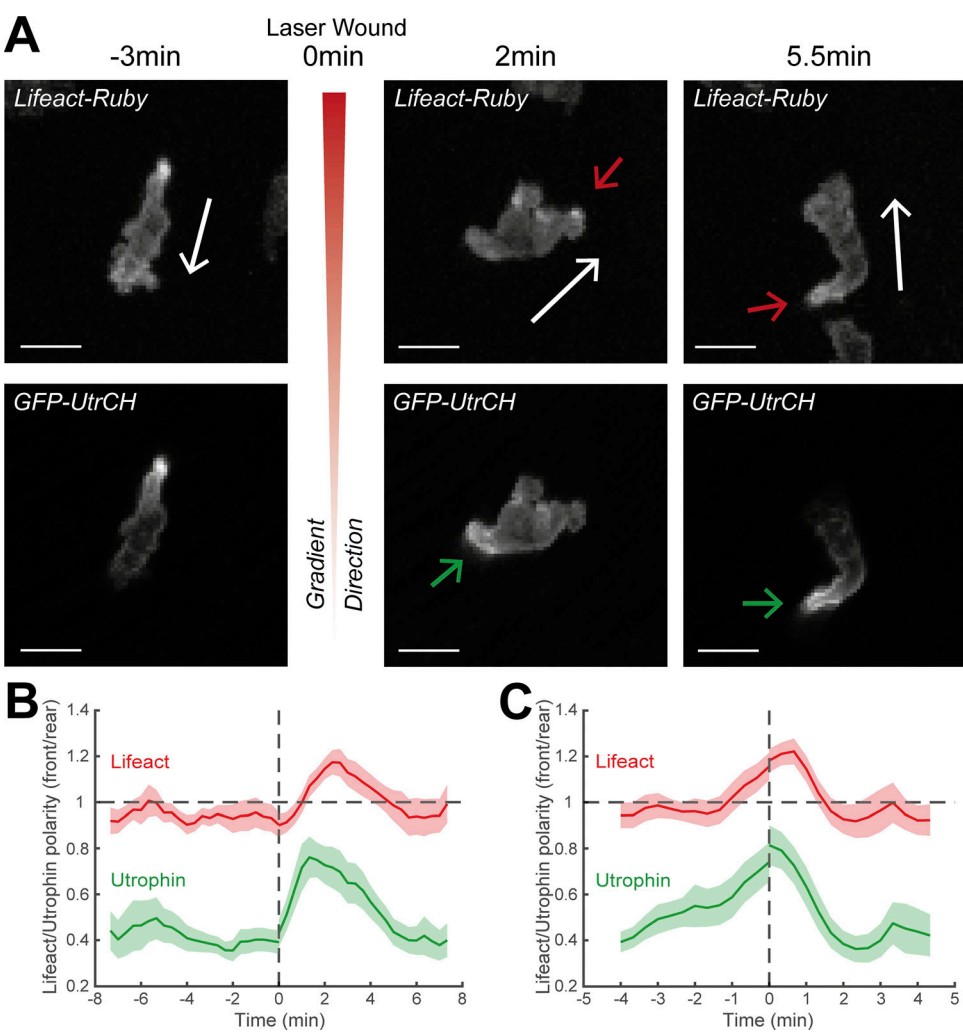

Figure S3. **Utrophin marker distribution during neutrophil gradient responses.** In support of Fig. 3. **(A)** Time-lapse sequence of two-photon confocal image projections showing a neutrophil (white) in a *Tg(mpx:Lifeact-Ruby)* zebrafish larva (top row), injected with GFP-UtrCH (bottom row), migrating pre- and post-LW. White arrows indicate the speed vector. Red/green arrows indicate the side of cell with higher Lifeact/Utrophin abundance. Scale bar = 10 µm. **(B and C)** Neutrophil Lifeact and Utrophin polarity (front/rear) in relation to time; Lifeact-Ruby is shown in red and Utrophin is shown in green. Time sequence was synchronized based on time of the LW (B) or the time that each individual neutrophil beginning of movement post-LW (C). *n* = 15 cells from 13 larvae.

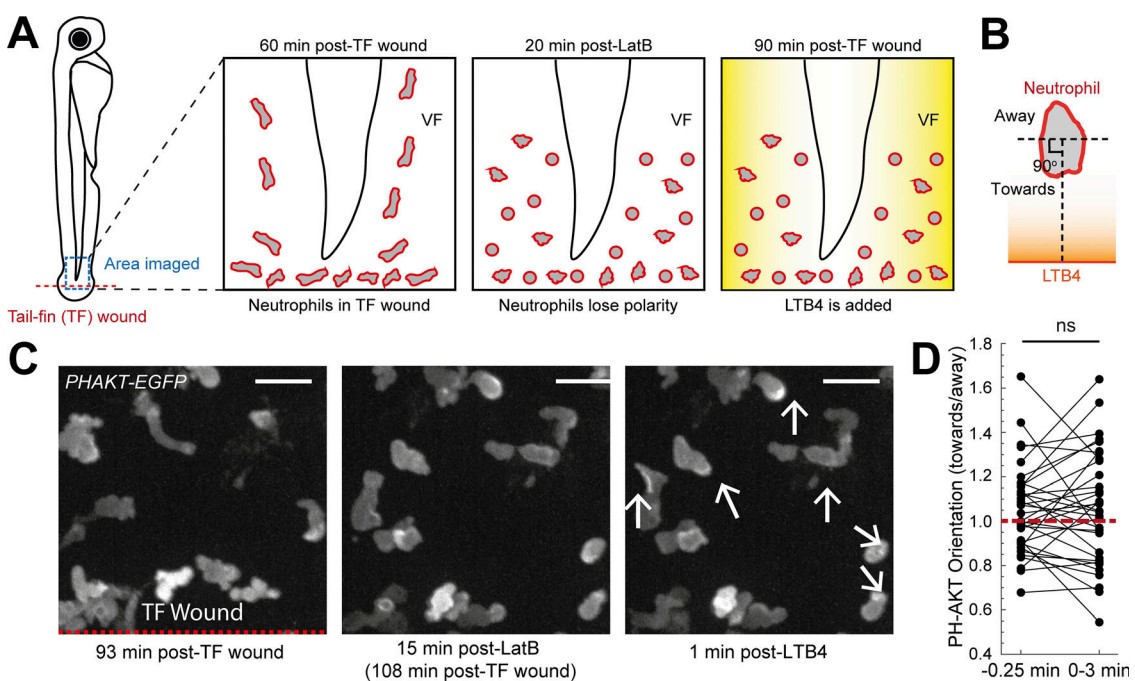

Figure S4. **Actin dynamics are required for gradient detection.** In support of Fig. 4. **(A)** Scheme of a zebrafish larva indicating the TF wound (red) and area of imaging (blue). Left: Neutrophils in a *Tg(mpx:PHAKT-EGFP)* infiltrating the TF. Middle: Neutrophils losing polarity and motility after addition of LatB. Right: Addition of LTB4 and assessment of signaling response. **(B)** Scheme that shows the separation of cell into two parts, one towards the LTB4 and one away from LTB4. The line that separates the cell is perpendicular to the line that connects the center of cell and the center of the boundary of the larva edge. **(C)** Left: Neutrophils in a *Tg(mpx:PHAKT-EGFP)* infiltrating the TF. Red dashed line indicates the TF wound. Middle: Neutrophils lose motility post-LatB addition. Right: Neutrophils (white arrows) immediately after LTB4 addition. Scale bar = 15 µm. **(D)** Plot of neutrophil PH-AKT orientation (towards/away), within a time window of 3 min after LTB4 addition. *n* = 34 cells from 5 larvae. Two-tailed paired *t* test.

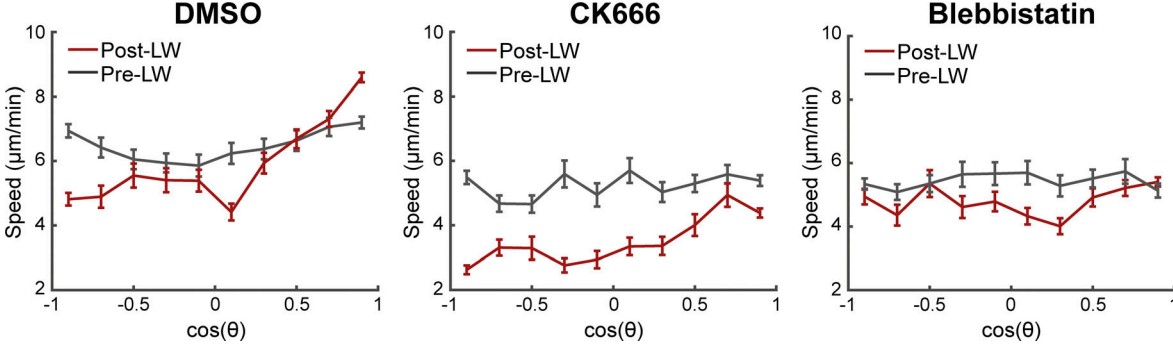

Figure S5. **Myosin-II inhibition impairs directional bias on speed.** In support of Fig. 5. Neutrophil speed in relation to the cosine θ for DMSO (left), CK666 (middle), and Blebbistatin (right), for pre- and post-LW. DMSO: *n* = 217–1,020 cell steps per bin pre-wound, *n* = 179–1,734 cell steps per bin post-wound (data from 9 larvae). CK666: *n* = 131–675 cell steps per bin pre-wound, *n* = 122–825 cell steps per bin post-wound (data from 5 larvae). Blebbistatin: *n* = 120–427 cell steps per bin pre-wound, *n* = 127–929 cell steps per bin post-wound (data from 4 larvae).

Video 1. **Neutrophil migration pre- and post-LW in a zebrafish larva.** In support of Figs. 1 and 2. Neutrophils in a *Tg(mpx:Lifeact-Ruby)* zebrafish larva in 30 nM LTB4. Time-lapse projections of Fig. 1 B correspond to 14:40, 24:40, 34:40, and 44:40 min at video. Frame interval is 20 s and frame rate is 15 fps. Scale bar = 50 µm.

Video 2. **Neutrophil migration pre- and post-LW in a mouse.** In support of Fig. 1. Migrating neutrophils in the ear dermis of C57Bl6-Albino mouse. LW occurs at 25 min, in the skin dermis. Time-lapse projections of Fig. S1 B correspond to 10, 25, and 55 min at video. Left: Second harmonic generation image (gray) with CTMPX-labeled neutrophils (red). Right: Neutrophil channel only. Frame interval is 30 s and frame rate is 15 fps. Scale bar = 50 µm.

Video 3. **Retrograde Myosin-II flows in zebrafish neutrophils migrating towards a wound.** In support of Fig. 2. **(A)** Neutrophil in a *Tg(actb1:myl12.1-eGFP)* zebrafish larva, migrating towards the TF wound. Red line denotes the segmented outline of the cell. White arrows indicate the velocity vector for each time-frame. **(B)** Myosin-II retrograde flows in the reference system of the cell. **(C)** Myosin-II retrograde flow speed vectors (red). **(D)** Heatmap of flow speeds. Frame interval is 1.93 s and frame rate is 6 fps. Scale bar = 10 μm.

Video 4. **Response of individual neutrophil to LW.** In support of Figs. 3 and S2. **(A)** Neutrophil in a *Tg(mpx:Lifeact-Ruby)* zebrafish larva in VF in 30 nM LTB4. LW occurs at 6:40 min (not visible in the video, location is indicated by angle). Time-lapse projections of Fig. 3 A correspond to 00:20, 07:40, 08:00, and 11:00 min, respectively, at video. Frame interval is 20 s. **(B)** Neutrophil in a *Tg(mpx:Lifeact-Ruby)* zebrafish larva in head. LW occurs at 5:00 min (not visible in the video, location is indicated by angle). Time-lapse projections of Fig. S2 C correspond to 4:30, 7, and 10 min, respectively, at video. Frame interval is 30 s. **(A and B)** White arrow indicates the neutrophil to observe. Red/green arrows indicate Lifeact at the front/rear. Frame rate is 5 fps. Scale bar = 20 μm.

Video 5. **Lifeact-Ruby and GFP-UtrCH distribution in a migrating neutrophil.** In support of Fig. S3. Neutrophil (white arrow) in a *Tg(mpx:Lifeact-Ruby)* zebrafish larva (left) in 30 nM LTB4, injected with *Tol2(mpx:GFP-UtrCH)* (right). Red/green arrows indicate high Lifeact-Ruby/GFP-UtrCH abundance. Time-lapse projections of Fig. S3 A correspond to 4, 9, and 12:20 min, respectively, at video. Frame interval is 20 s and frame rate is 15 fps. Scale bar = 20 μm.

Video 6. **PH-AKT distribution in immobilized cells before and after gradient exposure.** In support of Figs. 4 and S4. **(A)** Neutrophils in a *Tg(mpx:PHAKT-EGFP)* zebrafish larva initially migrating towards a TF wound (unspecific motion). Indicated time is in relation to TF wounding. Green arrows indicate PIP3 activity post-LTB4. Frame interval is 15 s and frame rate is 6 fps. Scale bar = 15 μm. **(B)** Neutrophils in a *Tg(mpx:PHAKT-EGFP)* zebrafish larva migrating randomly in the head and subsequently exposed to 1 μM LatB and LW. Arrows at 9 min indicate decelerating cells to observe. Arrows post-LW indicate cells with PIP3 activity. Frame interval is 30 s and frame rate is 6 fps. Scale bar = 20 μm. **(C)** Neutrophils in a *Tg(mpx:PHAKT-EGFP)* zebrafish larva migrating randomly in the head and subsequently exposed to and LW (no LatB). Arrow at 0 min indicates the cell to observe. Arrows at 6:40 min indicates the cell's PIP3 activity. Frame interval is 30 s and frame rate is 6 fps. Scale bar = 20 μm.

Video 7. **High dose of CK666 and Blebbistatin disrupts locomotion.** In support of Figs. 5 and S5. **(A and B)** Neutrophils in a *Tg(mpx:Lifeact-Ruby)* zebrafish larva in high dose of (A) CK666 (100 μM) and (B) Blebbistatin (200 μM). Frame interval is 20 s and frame rate is 15 fps. Scale bar = 30 μm.

Video 8. **Effects of inhibitors on leading-edge expansion.** In support of Figs. 5 and S5. **(A–D)** Examples of neutrophils (white arrows) in a *Tg(mpx:Lifeact-Ruby)* zebrafish larva in (A) DMSO, (B) 40–50 μM CK666, and (C and D) 150 μM Blebbistatin, pre- and post-LW. Time-lapse projections of Fig. 5 A (bottom row) correspond to 8:20, 11:40, and 14 min, respectively, at D. Red arrow in D shows the creation of the new front towards the source from the lateral/rear side. Frame interval is 20 s and frame rate is 5 fps. Scale bar = 20 μm.

