## [Peer Review File · The Journal of Cell Biology]

A two-step search and run response to gradients shapes leukocyte navigation in vivo

Antonios Georgantzoglou, Hugo Poplimont, Hazel Walker, Tim Lämmermann, and Milka Sarris

Corresponding Author(s): Milka Sarris, University of Cambridge

Review Timeline:

Submission Date:	2021-03-31
Editorial Decision:	2021-06-11
Revision Received:	2022-02-03
Editorial Decision:	2022-04-15
Revision Received:	2022-05-13

Monitoring Editor: Anna Huttenlocher

Scientific Editor: Dan Simon

Transaction Report:

DOI: <https://doi.org/10.1083/jcb.202103207>

June 11, 2021

Re: JCB manuscript #202103207

Dr. Milka Sarris
University of Cambridge
Physiology, Development and Neuroscience
Downing site
Cambridge CB2 3DY
United Kingdom

Dear Dr. Sarris,

Thank you for submitting your manuscript entitled "Interstitial leukocyte navigation through a search and run response to gradients." The manuscript was assessed by expert reviewers, whose comments are appended to this letter. We invite you to submit a revision if you can address the reviewers' key concerns, as outlined here.

1. Revise the text to more clearly support the data presented as suggested by several reviewers. Please also take care to clearly report what has previously been done and include appropriate citations as indicated by the reviewers. Barros-Becker et al have reported the effects of Arp2/3 inhibition on the motility of zebrafish neutrophils migrating in vivo and this should be cited.
2. As suggested by several reviewers GFP-actin or another probe should be used in this system to back up the reported findings.
3. Studies should be done without LTB4 and response to wounding alone.
4. Revisions suggested by reviewer #3 including moving some of the supplemental data to figure 1 should be done for clarity.
5. Data should be shown using dot plots rather than bar graphs.

When submitting the revision, please include a cover letter addressing all of the reviewers' comments point by point. Please also highlight all changes in the text of the manuscript.

GENERAL GUIDELINES:

Text limits: Character count for an Article is < 40,000, not including spaces. Count includes title page, abstract, introduction, results, discussion, acknowledgments, and figure legends. Count does not include materials and methods, references, tables, or supplemental legends.

Figures: Articles may have up to 10 main text figures. Figures must be prepared according to the policies outlined in our Instructions to Authors, under Data Presentation, <https://jcb.rupress.org/site/misc/ifora.xhtml>. All figures in accepted manuscripts will be screened prior to publication.

Supplemental information: There are strict limits on the allowable amount of supplemental data. Articles may have up to 5 supplemental figures. Up to 10 supplemental videos or flash animations are allowed. A summary of all supplemental material should appear at the end of the Materials and methods section.

As you may know, the typical timeframe for revisions is three to four months. However, we at JCB realize that the implementation of social distancing and shelter in place measures that limit spread of COVID-19 also pose challenges to scientific researchers. Lab closures especially are preventing scientists from conducting experiments to further their research. Therefore, JCB has waived the revision time limit. We recommend that you reach out to the editors once your lab has reopened to decide on an appropriate time frame for resubmission. Please note that papers are generally considered through only one revision cycle, so any revised manuscript will likely be either accepted or rejected.

Thank you for this interesting contribution to Journal of Cell Biology. You can contact us at the journal office with any questions, cellbio@rockefeller.edu or call (212) 327-8588.

Sincerely,

Anna Huttenlocher, MD
Monitoring Editor
Journal of Cell Biology

Dan Simon, PhD
Scientific Editor
Journal of Cell Biology

Reviewer #1 (Comments to the Authors (Required)):

The manuscript "Interstitial leukocyte navigation through a search and run response to gradients" by Georgantzoglou et al reports on the molecular mechanisms by which migrating cells (neutrophils in this case) respond to gradients of chemoattractants. The authors suggest a two-step process that leads to changes in the migration direction, such that the cells accurately move in the direction of the source of the attractant.

Overall, the general topic is of interest for the readership of the Journal and in general the technical level / clarity of the results is high.

The major issue the authors should be more convincing about is the degree of novelty, or advance presented in the work considering the previously published work. Some of the principles presented in this work were previously experimentally demonstrated for example by the Gerisch lab (Lange et al 2016), although not in vivo, with a different cell type and attractant, and not to such high level of measurements. Other issues listed below relate to the fact that in some cases the results described in the text are difficult to observe in the presented data. It could be that the text should be modified to make one see in the figures what the authors "want" the reader to see, but as it is, it is sometimes not clear enough.

- In Fig 1C, the directionality at 40 micrometers and 160 is similar, but the authors interpret the findings as indicating "...a marked increase in orientation bias across a distance range as far as 160 μ m from the wound perimeter". It seems to me like there is one point (20) where an increase is observed and that the conclusion depends on it. Also, if one compares the persistent migration level between 160 and 80, it is decreased, which is not in line with the conclusion. Could it be that the persistence pattern reflects tissue properties around the point where the wound is induced (pre-LW cells show the same profile), with post wound cells being in general more directed? This issue is not clear in the mouse case, where persistence increases also in the Pre-LW case.

- In the "Fast persistent motion is associated with rear enrichment of actin polymerisation probes and fast actin flows" section, the authors should make it clear what the difference between their findings and previously published material. Mention the theory Maiuri paper where the UCSP model was formulated and the findings presented there and compare them with the results presented here. The result from a relevant paper should perhaps be mentioned and compared with the results presented here (Ruprecht et al 2015).

- When measuring the actin flow it would be more convincing if a more direct label of actin is used, and in this context it is especially not clear why use Myosin-II-GFP.

- In the "Directional turns during the first phase of chemotaxis involve a searching signalling pattern" section, the authors state that "...establish this for the first time in a live tissue context, we assessed the response of the signalling mediator PI3K after inhibition of actin polymerisation with Latrunculin-B (LatB)". The result of this experiment was polarization PIP3 in random directions. The authors conclude that "in the absence of actin dynamics, gradients induce a searching signalling pattern and that directional decisions during the first phase of chemotaxis rely on active exploration.". Unless I missed it somehow, the initial PIP3 distribution after LTB4 / LW should be examined also in cells where Actin polymerization is not inhibited, to determine if the initial response is similar and the "correction" in the distribution of PIP3 occurs later, or whether it is initially oriented properly. This would support the idea that actin polymerization is essential for the initial response, or alternatively if it allows for corrections / stabilization of an initial slightly higher level of PIP3.

- It is not clear to me why state that the mean intensity of PHAKT-EGFP is not increased- How would the total signal of PHAKT-EGFP change in such an experiment?

- In the discussion the authors present what is perhaps the main novel conclusion of the paper. "When a wandering cell experiences a new chemical gradient, the first response is to stop and explore ('searching phase') rather than steer or accelerate." "search and run" phases were also suggested in the case of zebrafish germ cells, so it would be interesting to compare the two models as well. Could the "searching phase" be just a reflection of reaction to the initial relatively high chemoattractant (as compared with the pre-wound situation) all around the cell, with the cell gradually establishing a stable

front? If the cell exits the "search mode" towards a wound and a new wound is induced in the same orientation, would it stop as well? A stop would be consistent with a simple reaction to the high level of the chemical cue in the rear.

- In the "Differential roles of protrusive and contractile forces in search and run response to gradients" section the authors found that "while control DMSO-treated and blebbistatin-treated cells showed front actin enrichment and oriented turns, these events were suppressed in CK666-treated cells", which was "accompanied by suppressed turning during this phase, as the cosine of theta showed limited increase after gradient exposure in comparison to DMSO or blebbistatin-treated exposed cells". These results are not clearly visible in Figure 5 D - F. In Figure 5 E for example, the graph shows an increase in actin polarity after LW, as well as an increase in the cos (θ) value.

Minor points

1. I would exchange "To move appropriately to functional destination, cells" with something like "To reach their destination, migrating cells...".
2. I would remove the "swiftly" from "...to substrate to swiftly squeeze through tissue..".
3. Would be helpful if the time stamp on the videos corresponds to that in the Figures where snapshots from the movies are presented.
4. Define VF the first time "ventral fin" is mentioned and in the legend of Figure 1. Use the abbreviation as you use CHT after defining it. Same for LW and WP.
5. In 1D and 1I, put the "high to low" scale between the upper and lower panel (move it down a bit), such that it does not look like the lower panels in 1D and 1I belong to 1E and 1J respectively. Do panels 1D and 1I add information one does not get from the other panels in Figure 1 (E and J)?
6. While Delta is defined in length in the text, Theta is mentioned in the same location without defining it.
7. How many cells were examined in 2D and 2E to make the point these panels suggest?
8. "However, the interpretation of rear Lifeact enrichment during fast phases of motion remained less clear.". Exchanging "interpretation" with "significance" would perhaps fit better.

Reviewer #2 (Comments to the Authors (Required)):

Most work in the directed migration field has focused on regulation of cell protrusions at the leading edge, and much of this has focused on cells stimulated *in vitro*. In this manuscript by Georgantzoglou et al., "Interstitial leukocyte navigation through a search and run response to gradients", the authors investigate neutrophil movement *in vivo* and analyse actin flows as well as overall actin polarity. The authors find that neutrophils have two distinct phases for directed migration following initiation of a wound. In the first search phase, speed decreases and actin polymerisation occurs at the front of cell. In the second 'run' phase, cells migrate faster and with more persistence toward the wound site. The authors identify the Arp2/3 complex to be necessary for the search phase and myosin contractility for the run phase.

Overall, this is an important area and approach which satisfies a need in the field for good quantitative analysis of actin dynamics during directed migration *in vivo*. I also welcome the analysis of actin flows, which have been less-studied than the typical focus of protrusion regulation. I think this manuscript contains the nucleus of what could be nice paper of broad interest to the readers of JCB. But I have some significant concerns for the current iteration of the paper.

Major concerns

1. By far my biggest concern relates to the authors' choice of probe for reading out actin polymer. Their core conclusions for actin polarity and actin flows require an accurate visualization of endogenous actin polymer. There are many possible choices for ways to read out actin polymer, each of which have their issues. The authors use LifeAct, a common choice in the field, as their sole actin probe. But this is a poor choice for this context in that this probe fails to read out the lamellipodial actin that is the dominant mode of actin organization at the neutrophil leading edge. High-quality 3D imaging of neutrophils with lattice light sheet imaging shows that LifeAct fails to enrich in lamellipodia (Fritz-Laylin et al, 2017). And the authors note that other studies have noted that convection depletes Lifeact from the tips of lamellipodia that have fast flow (Yamashiro et al., 2019). This raises significant doubts on whether the authors observation of posterior enrichment of LifeAct really represents a shift in actin polymer toward the tail, or this probe is simply poor at recognizing the leading edge lamellipodial actin. Similarly, the enrichment of the LifeAct probe in the spreading phase could either be due to more actin polymerized in the front or a different organization of actin polymer or alterations in flow that is more easily visualized by LifeAct. The authors need to complement their LifeAct observations with an actin probe that doesn't have this defect. GFP-Actin appears to be the probe that is best at recognizing the overall filamentous actin distribution in living neutrophils.

2. The model presented here relies heavily on the use of two drugs, CK666 to perturb actin branching and blebbistatin to affect myosin contractility. As with all pharmacological inhibitors, these drugs are not perfect, and the study would benefit by adding orthogonal inhibitors. For the back program in particular there are other good (possibly better) drugs that could be used including the ROCK inhibitor Y27632 and the Rho inhibitor C3. These perturbations could also have the advantage in determining if other aspects of the back program besides myosin are participating in cell guidance, actin flows, etc. Genetic perturbations of these pathways would be even better, but I am not sure how feasible these are in a reasonable time frame in an *in vivo* setting.

3. Data presented in Figure S2 are not convincing. Significance seems driven mostly by a couple of data points. Untreated cells need to be included as a control to have a sense for how well their PIP3 polarity can be visualized in this setting. These experiments are mainly used to say that the exploration phase is necessary for gradient interpretation, but obviously latrunculin treatment affects many other cell processes beyond exploration (like cell mechanics and any other behaviors that depend on actin polymer or the cell cortex). The authors would need to do much more to support their claims than depolymerize the actin cytoskeleton such as other less extreme perturbations that alter cell exploration. Or they should scrap this figure.

More minor concerns

1. I understand why the authors add LTB4 prior to cell wounding, but the application of external gradients loses some of the beauty of studying a native *in vivo* system. LTB4 could alter cell motility, gradient interpretation, actin flows, etc beyond the wound-mediated cues. The authors should verify their core conclusions in zebrafish not pre-treated with LTB4.

2. I'm concerned that non-expert readers will be confused by some of the quantitative metrics used here. In particular the compound metrics that combine both delta and theta like Fig 1E. They should include schematics along with the main figures to explain these metrics.

3. How do the authors evaluate flows in cells with more complex multiple flows? In other words, for Figure 3, how do you distinguish between global reduction in flow and local change in flow directionality, for example when a cell turns and is no longer straight? This relates to the interesting point in the discussion on whether convergence of flows is an important regulator of actin behavior and cell movement. A little further analysis that doesn't just reduce the flows to a single number would be helpful here.

4. Figure 3 would greatly benefit to have no wound control. Although pre-wound controls are present in figure 4, it would be interesting to also have them in figure 3, especially when looking at myosin flow and the correlation between migration and myosin flow speed.

5. Line 276-277: This can be a bit misleading as the authors do not directly look at mechanics but rather affect pathways known to correlate with effects on mechanics. Needs to be rephrased.

Reference: Fritz-Laylin, Lillian K., et al. "Actin-based protrusions of migrating neutrophils are intrinsically lamellar and facilitate direction changes." *Elife* 6 (2017): e26990.

Reviewer #3 (Comments to the Authors (Required)):

How cells sense and respond to chemotactic gradients is a major area of research, and previous studies have identified roles for Arp 2/3-mediated protrusions towards a chemotactic gradient in direction sensing, though Arp2/3 is not required for directed motion in some cellular contexts. Emerging evidence suggests actin flow may help maintain polarity and respond to chemotactic gradients. This paper tries to distinguish roles of Arp2/3/protrusions versus actin flows in sensing and moving towards the direction of chemotactic gradients. They mainly use neutrophils migrating to a laser wound in the zebrafish ventral fin as a model but also show similar mechanisms occur in mouse neutrophils migrating in the ear dermis wound. They find that neutrophils respond to gradients by making small angled turns. With automated quantification of cell speed and actin polarization, the authors find that there are two phases for neutrophil migration: a slow phase where actin is polarized at the front of the cell and a fast phase where actin is polarized at the rear of the cell. They find that Arp 2/3 inhibition disrupts small turning of neutrophils while myosin inhibition reduces track straightness. They propose a model where neutrophils first search with small turns and protrusions at the front of the cell followed by a run phase where actin flows mediate persistent direction. This is an interesting and challenging study of cell behavior *in vivo*, which reports novel findings that will be of interest to JCB readers.

While the analysis and quantification are detailed, the graphs are not straightforward to interpret and figures are lacking key labels and explanations that would help the reader follow along. In addition, there are some questions remaining with the proposed model and methods that should be addressed prior to publication.

1. A reader must understand the experimental setup and what angles are measured to understand this paper. Therefore, it is essential to move this information from figure S1A,B and Fig. 2A into Figure 1. If necessary, to make space for this critical information, the mouse data could go into a separate figure or even into the supplement since it is confirmatory and not the main focus of the paper.

2. In Figure 1C, why is the trend the same for pre and post LW (goes down and then up)

3. Why report the cosine of angles rather than the actual angles in degrees? It would be simpler to report angles and more intuitive for the reader to interpret. If it is necessary for some reason to report the cosines, please be sure to label graphs to indicate what the cosines of angles mean in terms of direction, both in the figures and within the text (sometimes it is described but sometimes not)

4. Abbreviations (e.g. LW) are not defined in Figure 1 but are elsewhere and should be clearly defined throughout.
5. Please indicate directly on the figure what fluorescent marker is labeled (i.e., F-actin) so a reader does not have to hunt in the figure legend for the information.
6. Please replace bar graphs with more informative representations such as dot plots (when there are a few data points) or violin plots (when there are many points) or box and whisker plots to show the full distribution and variance within the data sets.
7. The authors discuss the two phases in Figure 2 but do not really quantify/demonstrate these phases until Figure 4- some reorganization of figure order to have 2 and 4 follow one another would make the paper easier to read. Also, how consistent are the two phases and the speeds that characterize the two phases
8. Can the authors describe more clearly how "track straightness" is calculated/ show how the turn angles change between the different phases more directly?
9. How does "beginning of motion" correspond to the slow and fast phases discussed?
10. In the search phase, are the authors proposing searching is through protrusions or other actin-based structures? Protrusions are discussed in the abstract and introduction but are not analyzed thoroughly in this paper.
11. It is unclear what particles are tracked with PIV in Figure 3 when the signal appears diffuse and lacking in particles
12. Why is Figure 3/actin flows not measured in reference to laser wounding or connected to the slow and fast phases more directly? Showing the data relative to LW timepoint would help clarify when changes in actin flow occur and more strongly support their model.
13. Can actin flows be tracked with other actin markers?
14. Not clear in Figure 4D what 3 minutes and 6 minutes refer to
15. Put figure S2 in main text since it seems to be a main point/should be included
16. Can the authors measure how each drug treatment affects the actin flows versus protrusions/branching of the neutrophils?
17. Figure 5, why does Arp2/3 inhibition not strongly inhibit direction sensing compared to blebbistatin? Calls into question the conclusion that the protrusions are important for steering the cell.

We thank all the reviewers for their thoughtful comments, which have assisted us in improving the manuscript substantially.

Reviewer #1: (Comments to the Authors (Required)):

The manuscript "Interstitial leukocyte navigation through a search and run response to gradients" by Georgantzoglou et al reports on the molecular mechanisms by which migrating cells (neutrophils in this case) respond to gradients of chemoattractants. The authors suggest a two-step process that leads to changes in the migration direction, such that the cells accurately move in the direction of the source of the attractant. Overall, the general topic is of interest for the readership of the Journal and in general the technical level / clarity of the results is high.

The major issue the authors should be more convincing about is the degree of novelty, or advance presented in the work considering the previously published work. Some of the principles presented in this work were previously experimentally demonstrated for example by the (Lange et Gerisch lab al 2016), although not in vivo, with a different cell type and attractant, and not to such high level of measurements. Other issues listed below relate to the fact that in some cases the results described in the text are difficult to observe in the presented data. It could be that the text should be modified to make one see in the figures what the authors "want" the reader to see, but as it is, it is sometimes not clear enough.

We agree with the reviewer on these points. We have performed alternative experiments/analyses in some cases where data were not so clear (drug treatments, PIP3 dynamics) and changed the text in the abstract, title and main body to better distinguish our findings from previous research. Based on new analyses with the inhibitors, we also highlight more the importance of contractility in gradient sensing, as this is not something that was evident by prior work. We cite Lange et al., 2016 in the introduction, where we make a case for why it is important to have experimental systems in vivo that capture real-time responses to gradient in physiological settings. Finally we highlight the importance of directly exploring gradient responses in vivo, since in vitro findings show different responses in different settings or sometimes do not predict behaviours of cells in vivo. We have changed the title slightly to emphasise the in vivo aspect of our study.

Major Points

1. In Fig 1C, the directionality at 40 micrometers and 160 is similar, but the authors interpret the findings as indicating "...a marked increase in orientation bias across a distance range as far as 160µm from the wound perimeter". It seems to me like there is one point (20) where an increase is observed and that the conclusion depends on it. Also, if one compares the persistent migration level between 160 and 80, it is decreased, which is not in line with the conclusion. Could it be that the persistence pattern reflects tissue properties around the point where the wound is induced (pre-LW cells show the same profile), with post wound cells being in general more directed? This issue is not clear in the mouse case, where persistence increases also in the Pre-LW case.

We have changed the text to clarify our description of these results.

Here we are referring to the average angle of the cells in relation to the source. This is clearly higher post-wound versus pre-wound across all distances, which means there is directional preference to the wound for cells across the whole range. The shape of the curve is a different matter. Indeed, the angle increases linearly from 100µm to 0µm (indicating that the gradient becomes stronger as cells approach) whereas the shape of the curve becomes anomalous beyond 100µm. Since this is unspecific to the presence of the gradient/wound, we indeed interpret this as tissue geometry constraints on cell angles. This is not surprising

as 100µm is the approximate width of the fin tissue. In the mouse sample the tissue is less anomalous and so the shape of the curve is more linear. We have added new text in the manuscript to help interpretation of this graph.

2. In the "Fast persistent motion is associated with rear enrichment of actin polymerisation probes and fast actin flows" section, the authors should make it clear what the difference between their findings and previously published material. Mention the theory Maiuri paper where the UCSP model was formulated and the findings presented there and compare them with the results presented here. The result from a relevant paper should perhaps be mentioned and compared with the results presented here (Ruprecht et al 2015).

We realise that the title of this result section and the organisation of figures 2 and 3 did not highlight the main point we want to make. We fused figure 2 and 3 and rephrased this result section to: 'Rear enrichment of actin polymerisation probes reports phases of fast actin flows'. We cite here and elsewhere both Mauri et al., 2015 and Ruprecht et al., 2015 and explain that our data in vivo are consistent with flow analyses they made in vitro. But the main point for us was to link Lifeact distribution with speed of flows because this is important for interpretation of rear actin enrichment subsequently in the paper. As we explain, it is difficult to directly measure the temporal profile of actin flows in vivo before and after gradient exposure, but we can measure temporal evolution of Lifeact distribution and from this infer the status of actin flows.

3. When measuring the actin flow it would be more convincing if a more direct label of actin is used, and in this context it is especially not clear why use Myosin-II-GFP.

It is not uncommon to use Myosin-based reported for flows. In Mauri et al., 2015 and Ruprecht et al. 2015 myosin-light-chain probe has been used and reveals the flows more clearly than actin probes. In our system, we could not discern flows with Lifeact, which binds too briefly, nor Utrophin, which binds too stably, perhaps due to the high speed of motion of these cells.

4. In the "Directional turns during the first phase of chemotaxis involve a searching signalling pattern" section, the authors state that "...establish this for the first time in a live tissue context, we assessed the response of the signalling mediator PI3K after inhibition of actin polymerisation with Latrunculin-B (LatB)". The result of this experiment was polarization PIP3 in random directions. The authors conclude that "in the absence of actin dynamics, gradients induce a searching signalling pattern and that directional decisions during the first phase of chemotaxis rely on active exploration.". Unless I missed it somehow, the initial PIP3 distribution after LTB4 / LW should be examined also in cells where Actin polymerization is not inhibited, to determine if the initial response is similar and the "correction" in the distribution of PIP3 occurs later, or whether it is initially oriented properly. This would support the idea that actin polymerization is essential for the initial response, or alternatively if it allows for corrections / stabilization of an initial slightly higher level of PIP3.

We agree with the reviewer and we have now included these datasets as a main figure. We removed the other metrics following comments by other reviewers and just focused on the simple metric of reorientation of PIP3 polarity (ratio of PIP3 intensity front/back). In cells with intact actin dynamics, PIP3 polarity is enhanced or reoriented whereas in the absence of actin dynamics cells fail to adjust PIP3 polarity along the gradient.

5. It is not clear to me why state that the mean intensity of PHAKT-EGFP is not increased- How would the total signal of PHAKT-EGFP change in such an experiment?

Here we were referring to artificial fluctuations in intensity from the microscopy instrumentation. In any case, we have now removed this plot because we believe the clearest metric is the polarity of PIP3 and the rest of the metrics added confusion.

6. In the discussion the authors present what is perhaps the main novel conclusion of the paper. "When a wandering cell experiences a new chemical gradient, the first response is to stop and explore ('searching phase') rather than steer or accelerate." "search and run" phases were also suggested in the case of zebrafish germ cells, so it would be interesting to compare the two models as well. Could the "searching phase" be just a reflection of reaction to the initial relatively high chemoattractant (as compared with the pre-wound situation) all around the cell, with the cell gradually establishing a stable front? If the cell exits the "search mode" towards a wound and a new wound is induced in the same orientation, would it stop as well? A stop would be consistent with a simple reaction to the high level of the chemical cue in the rear.

We agree this is an interesting mechanism to explore but these experiments are not so straightforward and beyond the scope of the paper. However, we added discussion of this idea in the discussion section.

We do not think this observation is the only novelty. In the revised paper, we hope that several points of novelty come across:

-We reveal the time-resolved sequence of changes in cells as they respond to gradients in vivo, which revealed a two-stage process including a previously undocumented deceleration stage.

-We show that pure spatial sensing without actin dynamics and exploration is insufficient for navigation in vivo

-We reveal the different contributions of protrusions and actin flows in the gradient sensing process and provide important evidence that contractility provides memory in gradient detection by stabilising polarity along gradients. In our opinion, the latter is the most important novelty because it provides a fundamental mechanistic element for temporal sensing and explains how cells would sense gradients without diversification of the leading edge.

7. In the "Differential roles of protrusive and contractile forces in search and run response to gradients" section the authors found that "while control DMSO-treated and blebbistatin-treated cells showed front actin enrichment and oriented turns, these events were suppressed in CK666-treated cells", which was "accompanied by suppressed turning during this phase, as the cosine of theta showed limited increase after gradient exposure in comparison to DMSO or blebbistatin-treated exposed cells". These results are not clearly visible in Figure 5 D - F. In Figure 5 E for example, the graph shows an increase in actin polarity after LW, as well as an increase in the cos (θ) value.

We agree with the reviewer and revised the analysis on this part of the paper. We realised that comparing time plots is not very meaningful as cells under drug treatment are slower in responding and don't show sufficient synchrony. Instead of these plots, we considered it would be more meaningful to show the cellular phenotypes caused by the treatments and link them to chemotaxis defects.

We now provide these new datasets to describe the cellular phenotypes:

-data showing that CK666 treatment reduces leading edge surface as expected. This supports the role of Arp2/3 in exploration.

-data showing that Blebbistatin treatment increases propensity to form entirely new front. This supports the idea that contractility provides memory in polarity state.

-data showing that Blebbistatin treatment inhibits the running phase but increases the search phase after gradient exposure. This supports the idea that biasing speed of actin flows in the gradient direction requires contractility.

-we revised the images to illustrate these cellular phenotypes

We then link these cellular phenotypes to cell motion defects:

-CK666-treated cells have specific defect in making correct turns (this was already in previous version of the paper)

-Blebbistatin-treated cells have specific defect in biasing cell speed according to direction (this was already in previous version of the paper in supplementary figure but we add also a simplified bar graph to illustrate this defect in the main figure)

-Blebbistatin-treated and CK666-treated cells have different defects in persistence (this was already in the previous version of the paper). CK666 affects persistence in longer timescales, which is easy to interpret because wrong turns would cumulatively affect the shape of the trajectory over time. Blebbistatin however affects persistence in a more fundamental way in short and long timescales, which is consistent with the idea that cells cannot keep a short-term memory of gradient detection.

Minor points

8. I would exchange "To move appropriately to functional destination, cells" with something like "To reach their destination, migrating cells...".

We have done this

9. I would remove the "swiftly" from "..to substrate to swiftly squeeze through tissue..".

We have done this

10. Would be helpful if the time stamp on the videos corresponds to that in the Figures where snapshots from the movies are presented.

Ideally we would put a timestamp with the point of laser wounding as zero in both cases. But we could not put negative values in the timestamp of the videos. To help readers relate the two, we added a comment in the video legends on how the timepoints correspond to the timepoints in the figure images.

11. Define VF the first time "ventral fin" is mentioned and in the legend of Figure 1. Use the abbreviation as you use CHT after defining it. Same for LW and WP.

We have defined the abbreviations as described

12. In 1D and 1I, put the "high to low" scale between the upper and lower panel (move it down a bit), such that it does not look like the lower panels in 1D and 1I belong to 1E and 1J respectively. Do panels 1D and 1I add information one does not get from the other panels in Figure 1 (E and J)?

The colour scale was moved lower as suggested.

13. While Delta is defined in length in the text, Theta is mentioned in the same location without defining it.

We added a further explanation of angle theta.

14. How many cells were examined in 2D and 2E to make the point these panels suggest?

Each plot (Figure S2A and S2B in current version) represents one cell, we reported these plots as example plots of two cells. To extract the correlation plots, we examined 21 migrating cells from 9 zebrafish larvae. In the revised version of the paper we moved the example plots to supplementary figure 2, in the process of compressing with figure 3.

15. "However, the interpretation of rear Lifeact enrichment during fast phases of motion remained less clear.". Exchanging "interpretation" with "significance" would perhaps fit better.

We amended this wording.

Reviewer #2: (Comments to the Authors (Required)):

Most work in the directed migration field has focused on regulation of cell protrusions at the leading edge, and much of this has focused on cells stimulated in vitro. In this manuscript by Georgantzoglou et al., "Interstitial leukocyte navigation through a search and run response to gradients", the authors investigate neutrophil movement in vivo and analyse actin flows as well as overall actin polarity. The authors find that neutrophils have two distinct phases for directed migration following initiation of a wound. In the first search phase, speed decreases and actin polymerisation occurs at the front of cell. In the second 'run' phase, cells migrate faster and with more persistence toward the wound site. The authors identify the Arp2/3 complex to be necessary for the search phase and myosin contractility for the run phase.

Overall, this is an important area and approach which satisfies a need in the field for good quantitative analysis of actin dynamics during directed migration in vivo. I also welcome the analysis of actin flows, which have been less-studied than the typical focus of protrusion regulation. I think this manuscript contains the nucleus of what could be nice paper of broad interest to the readers of JCB. But I have some significant concerns for the current iteration of the paper.

Major concerns

1. By far my biggest concern relates to the authors' choice of probe for reading out actin polymer. Their core conclusions for actin polarity and actin flows require an accurate visualization of endogenous actin polymer. There are many possible choices for ways to read out actin polymer, each of which have their issues. The authors use LifeAct, a common choice in the field, as their sole actin probe. But this is a poor choice for this context in that this probe fails to read out the lamellipodial actin that is the dominant mode of actin organization at the neutrophil leading edge. High-quality 3D imaging of neutrophils with lattice light sheet imaging shows that LifeAct fails to enrich in lamellipodia (Fritz-Laylin et al, 2017). And the authors note that other studies have noted that convection depletes Lifeact from the tips of lamellipodia that have fast flow (Yamashiro et al., 2019). This raises significant doubts on whether the authors observation of posterior enrichment of LifeAct really represents a shift in actin polymer toward the tail, or this probe is simply poor at recognizing the leading edge lamellipodial actin. Similarly, the enrichment of the LifeAct probe in the spreading phase could either be due to more actin polymerized in the front or a different organization of actin polymer or alterations in flow that is more easily visualized by LifeAct. The authors need to complement their LifeAct observations with an actin probe that doesn't have this defect. GFP-Actin appears to be the probe that is best at recognizing the overall filamentous actin distribution in living neutrophils.

Our interpretation of rear actin accumulation during fast phases of motion is that the probe may be transported to the rear by fast actin flows (as suggested by Yamashiro et al., 2019) rather than reflect new polymer at the rear. This is backed by analysis showing that the faster the cells are, the faster the flows and the more accumulation of Lifeact is observed at the rear. We make use of rear Lifeact distribution as a proxy for phases of fast actin flows in

the cells rather than make claims about the location of actin polymerisation during these phases.

To corroborate these findings with another probe, we considered which probe has been used successfully in zebrafish neutrophil before as there can be developmental effects in such genetic experiments. The main probes validated in fish neutrophils are Lifeact, which labels dynamic actin and Utrophin, which labels more stable networks but there are no reports of GFP-actin. We added analysis with the Utrophin probe (also used in Fritz-Laylin et al., 2017). Even though Utrophin accumulates at the rear and is largely excluded from the front, we see that in the first phase it accumulates less at the rear than during the running phase. This is consistent with the idea that these probes accumulate at the rear as a result of faster flows.

2. The model presented here relies heavily on the use of two drugs, CK666 to perturb actin branching and blebbistatin to affect myosin contractility. As with all pharmacological inhibitors, these drugs are not perfect, and the study would benefit by adding orthogonal inhibitors. For the back program in particular there are other good (possibly better) drugs that could be used including the ROCK inhibitor Y27632 and the Rho inhibitor C3. These perturbations could also have the advantage in determining if other aspects of the back program besides myosin are participating in cell guidance, actin flows, etc. Genetic perturbations of these pathways would be even better, but I am not sure how feasible these are in a reasonable time frame in an in vivo setting.

We agree that it would be ideal to have a range of inhibitors but these experiments are time consuming in vivo as it requires titration of many concentrations to find intermediate doses that do not prohibit motion but allow detection of effects on gradient sensing (we have now added a relevant comment and supplementary video regarding the concentration used). For this reason, we felt we had to prioritise other experiments that were required for the core conclusions. If this comment arises because the data with the two drugs were not clear enough, we have now significantly improved our analysis to more clearly show the effects.

3. Data presented in Figure S2 are not convincing. Significance seems driven mostly by a couple of data points. Untreated cells need to be included as a control to have a sense for how well their PIP3 polarity can be visualized in this setting. These experiments are mainly used to say that the exploration phase is necessary for gradient interpretation, but obviously latrunculin treatment affects many other cell processes beyond exploration (like cell mechanics and any other behaviors that depend on actin polymer or the cell cortex). The authors would need to do much more to support their claims than depolymerize the actin cytoskeleton such as other less extreme perturbations that alter cell exploration. Or they should scrap this figure.

Since there were varying recommendations from reviewers on this part, our approach was to strengthen the results with the controls suggested and add these data as a main figure.

We have done the following:

-We removed the metrics of contrast and intensity which do not show big changes as indicated by this reviewer.

-We kept the PIP3 polarity as a simpler measure and added new data from non-treated cells as comparison. While non-treated cells adjusted PIP3 polarity (either by shift in polarity or enhancement) the treated cells did not alter PIP3 distribution in response to gradients.

-We now show at least two ways via which actin dynamics contribute to gradient sensing. We measure the leading edge surface area in CK666-treated cells and show that this is affected. More importantly, we show that contractility is important for memory because blebbistatin-treated cells form an entirely new front in response to the gradient more easily than control cells but they fail to stabilise polarity along the gradient and get stuck in search mode. This memory provides at least one major mechanism that would explain why cells with perturbed actin dynamics cannot resolve the gradient in a purely spatial manner.

We do not exclude other ways that actin dynamics contribute to gradient sensing (e.g. mechanical feedback) beyond the two mechanisms we highlight in this paper. But the fact that some form of feedback from actin dynamics is needed in vivo, is an important observation in itself.

More minor concerns

4. I understand why the authors add LTB4 prior to cell wounding, but the application of external gradients loses some of the beauty of study a native in vivo system. LTB4 could alter cell motility, gradient interpretation, actin flows, etc beyond the wound-mediated cues. The authors should verify their core conclusions in zebrafish not pre-treated with LTB4.

To address this concern, we performed live imaging and laser wounding in a different location where neutrophils show constitutive motility without exogenous signals (Supplementary Figure 2). We find the same sequence of Lifeact and motion responses as described in Figure 3 for the LTB4-fin laser wound assay.

5. I'm concerned that non-expert readers will be confused by some of the quantitative metrics used here. In particular the compound metrics that combine both delta and theta like Fig 1E. They should include schematics along with the main figures to explain these metrics.

We appreciate this point and included the schematics in the main figures. We generally tried to simplify our data particularly in the last part with the drug treatments, where we now show bar graphs that are easier to interpret.

6. How do the authors evaluate flows in cells with more complex multiple flows? In other words, for Figure 3, how do you distinguish between global reduction in flow and local change in flow directionality, for example when a cell turns and is no longer straight? This relates to the interesting point in the discussion on whether convergence of flows is an important regulator of actin behavior and cell movement. A little further analysis that doesn't just reduce the flows to a single number would be helpful here.

We measure average rearward flow speed, which would be higher when flows in different parts of the cell are fast in the same direction. We do not specifically investigate sub-species of flows. Such analysis has been done in drosophila hemocytes or fish keratinocytes which have big lamella and more data can be acquired regarding subspecies of flows in the cells. For neutrophils in vivo with their less expanded shape, one would need more cells and such analyses could form a basis of future studies.

The main point of our actin flow analyses is to relate actin flow speed with cell behaviour and rear actin enrichment in our system so that we can make some basic inferences on temporal changes in actin flow status during the more complex gradient assays.

7. Figure 3 would greatly benefit to have no wound control. Although pre-wound controls are present in figure 4, it would be interesting to also have them in figure 3, especially when looking at myosin flow and the correlation between migration and myosin flow speed.

We agree this would be useful but we explain here and now also in the paper why this is technically difficult.

-The key assay for capturing gradient responses is the use of laser wounding as it allows monitoring cells right before and after. We do not have laser wounding coupled to a spinning disk scope and the two photon microscope does not provide us with sufficient temporal and spatial resolution for actin flow analysis.

- Even if we had such set up, it is difficult to provide temporal profiles for actin flows, as it requires actin flow to be tracked in all cells for a consistent timescale before and after wounding - this is limited in these 3D environments because the imaging volume has to be small in order to image at high temporal resolution and cells exit the imaging volume at variable time scales.

For this reason, we used rear enrichment of actin as indirect indicator of phases of fast actin flows. We hope the rationale is more clear now.

8. Line 276-277: This can be a bit misleading as the authors do not directly look at mechanics but rather affect pathways known to correlate with effects on mechanics. Needs to be rephrased.

We understand this refers to the phrasing 'protrusive and contractile' forces in the result title. We have rephrased this section to: 'Differential roles of protrusive and contractile actin structures in gradient sensing' and removed the word 'forces'.

Reviewer #3: (Comments to the Authors (Required)):

How cells sense and respond to chemotactic gradients is a major area of research, and previous studies have identified roles for Arp 2/3-mediated protrusions towards a chemotactic gradient in direction sensing, though Arp2/3 is not required for directed motion in some cellular contexts. Emerging evidence suggests actin flow may help maintain polarity and respond to chemotactic gradients. This paper tries to distinguish roles of Arp2/3/protrusions versus actin flows in sensing and moving towards the direction of chemotactic gradients. They mainly use neutrophils migrating to a laser wound in the zebrafish ventral fin as a model but also show similar mechanisms occur in mouse neutrophils migrating in the ear dermis wound. They find that neutrophils respond to gradients by making small angled turns. With automated quantification of cell speed and actin polarization, the authors find that there are two phases for neutrophil migration: a slow phase where actin is polarized at the front of the cell and a fast phase where actin is polarized at the rear of the cell. They find that Arp 2/3 inhibition disrupts small turning of neutrophils while myosin inhibition reduces track straightness. They propose a model where neutrophils first search with small turns and protrusions at the front of the cell followed by a run phase where actin flows mediate persistent direction. This is an interesting and challenging study of cell behavior in vivo, which reports novel findings that will be of interest to JCB readers.

While the analysis and quantification are detailed, the graphs are not straightforward to interpret and figures are lacking key labels and explanations that would help the reader follow along. In addition, there are some questions remaining with the proposed model and methods that should be addressed prior to publication.

Points

1. A reader must understand the experimental setup and what angles are measured to understand this paper. Therefore, it is essential to move this information from figure S1A,B and Fig. 2A into Figure 1. If necessary, to make space for this critical information, the mouse data could go into a separate figure or even into the supplement since it is confirmatory and not the main focus of the paper.

We have performed these changes.

2. In Figure 1C, why is the trend the same for pre and post LW (goes down and then up)

Please see answer to reviewer 1 point 1. We have added new text in the manuscript to help interpretation of this graph.

3. Why report the cosine of angles rather than the actual angles in degrees? It would be simpler to report angles and more intuitive for the reader to interpret. If it is necessary for some reason to report the cosines, please be sure to label graphs to indicate what the cosines of angles mean in terms of direction, both in the figures and within the text (sometimes it is described but sometimes not)

The cosine is often preferred (e.g. Weber et al., 2013, Lämmermann et al., 2013), because values are between -1 and 1, making it easier to perform computations and correlation analyses. To facilitate interpretation of cosines, we added an explanation of what cosine of theta closer to 1 or -1 means and what cosine of delta closer to 1 or -1 means.

4. Abbreviations (e.g. LW) are not defined in Figure 1 but are elsewhere and should be clearly defined throughout.

We made sure abbreviations are defined in the text and in legend of Figure 1 when they appeared for first time.

5. Please indicate directly on the figure what fluorescent marker is labeled (i.e., F-actin) so a reader does not have to hunt in the figure legend for the information.

Indication of the fluorescent marker was added to all figures. In figures where a series of time-lapse images is presented, the marker is indicated in the first time-point only.

6. Please replace bar graphs with more informative representations such as dot plots (when there are a few data points) or violin plots (when there are many points) or box and whisker plots to show the full distribution and variance within the data sets.

All bar-plots were replaced by dot-plots which show the SEM, with the exception of figure 5F where the amount of data points is too high.

7. The authors discuss the two phases in Figure 2 but do not really quantify/demonstrate these phases until Figure 4- some reorganization of figure order to have 2 and 4 follow one another would make the paper easier to read. Also, how consistent are the two phases and the speeds that characterize the two phases.

We have reorganised figures 2 and 3, in new figure 2, which aims to provide an interpretation for the accumulation of Lifeact in relation to cell and actin flow speed. We do not refer to phases until new figure 3 (previous figure 4). We hope this is clearer now.

8. Can the authors describe more clearly how "track straightness" is calculated/ show how the turn angles change between the different phases more directly?

We added an explanation of how we calculated the track straightness in the Methods section (line 639-641).

9. How does "beginning of motion" correspond to the slow and fast phases discussed?

Beginning of motion is the stage when the cell starts to move and accelerate after laser wound. This is usually in the range of 3-5 minutes but is not entirely synchronised across different cells.

10. In the search phase, are the authors proposing searching is through protrusions or other actin-based structures? Protrusions are discussed in the abstract and introduction but are not analyzed thoroughly in this paper.

Indeed we did not provide direct analysis of protrusions in the previous version. To address this point we added an analysis of leading edge surface area with or without drug treatment. This showed that Arp2/3 inhibition reduced the surface of leading edge. We therefore now interpret that exploration is through expansion of actin networks and the leading edge surface.

11. It is unclear what particles are tracked with PIV in Figure 3 when the signal appears diffuse and lacking in particles

PIV divides computationally the cytoplasm into small areas that can have similar intensity distributions and move together. This is referred to as 'particles' but does not necessarily require discrete/high contrast structures, as long as there are sets of pixels with similar intensity moving together.

12. Why is Figure 3/actin flows not measured in reference to laser wounding or connected to the slow and fast phases more directly? Showing the data relative to LW timepoint would help clarify when changes in actin flow occur and more strongly support their model.

Please see answer to reviewer 2 point 7. We now added an explanation for this in the paper.

13. Can actin flows be tracked with other actin markers?

Please see answer to reviewer 1 point 3. We tried Lifeact and Utrophin and could not discern actin flows as clearly. As in Mauri et al., 2015 myosin-based probes seem to give clearer view of actin flows.

14. Not clear in Figure 4D what 3 minutes and 6 minutes refer to

It means we calculate track straightness for the motion of the cell in the first 3 or 6 minutes after beginning of movement (or 3 and 6 minutes immediately prior to movement). We added an explanation in the legend.

15. Put figure S2 in main text since it seems to be a main point/should be included

We added this as a main figure with data requested by other reviewers. This is now figure 4.

16. Can the authors measure how each drug treatment affects the actin flows versus protrusions/branching of the neutrophils?

We measured the surface area of the leading edge for this. As expected CK666 reduced the surface area.

17. Figure 5, why does Arp2/3 inhibition not strongly inhibit direction sensing compared to blebbistatin? Calls into question the conclusion that the protrusions are important for steering the cell.

We agree this is one of the most important points in the paper and provide more discussion on this. Our data suggest that contractility is more fundamental, since when this is perturbed cells get stuck in search mode and show more pronounced defects in persistence than when leading edge dynamics are restricted. On the other hand, the ability to expand the leading edge may have more context-dependent contributions in gradient sensing. One could imagine in complex 3D settings with multiple crossroads that the ability to make correct turns (driven by protrusions) would have important contribution, whereas in confined 1D migration settings cells would merely rely on adjusting speed and persistence (which are driven by contractility).

April 15, 2022

RE: JCB Manuscript #202103207R

Dr. Milka Sarris
University of Cambridge
Physiology, Development and Neuroscience
Downing site
Cambridge CB2 3DY
United Kingdom

Dear Dr. Sarris,

Thank you for submitting your revised manuscript entitled "A two-step search and run response to gradients shapes leukocyte navigation in vivo." Thank you also for your patience with the peer review process. We would be happy to publish your paper in JCB pending final text revisions necessary to address the remaining reviewer comments and also to meet our formatting guidelines (see details below). We do not believe that further experiments are necessary to answer the reviewer comments.

A. MANUSCRIPT ORGANIZATION AND FORMATTING:

- 1) Text limits: Character count for Articles is < 40,000, not including spaces. Count includes title page, abstract, introduction, results, discussion, and acknowledgments. Count does not include materials and methods, figure legends, references, tables, or supplemental legends.
- 2) Figures: Articles may have up to 10 main text figures. Scale bars must be present on all microscopy images, including inset magnifications. Molecular weight or nucleic acid size markers must be included on all gel electrophoresis.
- 3) Statistical analysis: Error bars on graphic representations of numerical data must be clearly described in the figure legend. The number of independent data points (n) represented in a graph must be indicated in the legend. Statistical methods should be explained in full in the materials and methods. For figures presenting pooled data the statistical measure should be defined in the figure legends. Please also be sure to indicate the statistical tests used in each of your experiments (both in the figure legend itself and in a separate methods section) as well as the parameters of the test (for example, if you ran a t-test, please indicate if it was one- or two-sided, etc.). Also, if you used parametric tests, please indicate if the data distribution was tested for normality (and if so, how). If not, you must state something to the effect that "Data distribution was assumed to be normal but this was not formally tested."
- 4) Materials and methods: Should be comprehensive and not simply reference a previous publication for details on how an experiment was performed. Please provide full descriptions (at least in brief) in the text for readers who may not have access to referenced manuscripts. The text should not refer to methods "...as previously described."
- 5) For all cell lines, vectors, constructs/cDNAs, etc. - all genetic material: please include database / vendor ID (e.g., Addgene, ATCC, etc.) or if unavailable, please briefly describe their basic genetic features, even if described in other published work or gifted to you by other investigators (and provide references where appropriate). Please be sure to provide the sequences for all of your oligos: primers, si/shRNA, RNAi, gRNAs, etc. in the materials and methods. You must also indicate in the methods the source, species, and catalog numbers/vendor identifiers (where appropriate) for all of your antibodies, including secondary. If antibodies are not commercial please add a reference citation if possible.
- 6) Microscope image acquisition: The following information must be provided about the acquisition and processing of images:
 - a. Make and model of microscope
 - b. Type, magnification, and numerical aperture of the objective lenses
 - c. Temperature
 - d. Imaging medium
 - e. Fluorochromes
 - f. Camera make and model
 - g. Acquisition software
 - h. Any software used for image processing subsequent to data acquisition. Please include details and types of operations

involved (e.g., type of deconvolution, 3D reconstitutions, surface or volume rendering, gamma adjustments, etc.).

7) References: There is no limit to the number of references cited in a manuscript. References should be cited parenthetically in the text by author and year of publication. Abbreviate the names of journals according to PubMed.

8) Supplemental materials: There are strict limits on the allowable amount of supplemental data. Articles may have up to 5 supplemental figures and 10 videos. Please also note that tables, like figures, should be provided as individual, editable files. A summary of all supplemental material should appear at the end of the Materials and methods section. Please include one brief sentence per item.

9) Video legends: Should describe what is being shown, the cell type or tissue being viewed (including relevant cell treatments, concentration and duration, or transfection), the imaging method (e.g., time-lapse epifluorescence microscopy), what each color represents, how often frames were collected, the frames/second display rate, and the number of any figure that has related video stills or images.

10) eTOC summary: A ~40-50 word summary that describes the context and significance of the findings for a general readership should be included on the title page. The statement should be written in the present tense and refer to the work in the third person. It should begin with "First author name(s) et al..." to match our preferred style.

11) Conflict of interest statement: JCB requires inclusion of a statement in the acknowledgements regarding competing financial interests. If no competing financial interests exist, please include the following statement: "The authors declare no competing financial interests." If competing interests are declared, please follow your statement of these competing interests with the following statement: "The authors declare no further competing financial interests."

12) A separate author contribution section is required following the Acknowledgments in all research manuscripts. All authors should be mentioned and designated by their first and middle initials and full surnames. We encourage use of the CRediT nomenclature (<https://casrai.org/credit/>).

13) ORCID IDs: ORCID IDs are unique identifiers allowing researchers to create a record of their various scholarly contributions in a single place. At resubmission of your final files, please consider providing an ORCID ID for as many contributing authors as possible.

14) Please note that JCB now requires authors to submit Source Data used to generate figures containing gels and Western blots with all revised manuscripts. This Source Data consists of fully uncropped and unprocessed images for each gel/blot displayed in the main and supplemental figures. If your paper contains cropped gel and/or blot images, please be sure to provide one Source Data file for each figure that contains gels and/or blots along with your revised manuscript files. File names for Source Data figures should be alphanumeric without any spaces or special characters (i.e., SourceDataF#, where F# refers to the associated main figure number or SourceDataFS# for those associated with Supplementary figures). The lanes of the gels/blots should be labeled as they are in the associated figure, the place where cropping was applied should be marked (with a box), and molecular weight/size standards should be labeled wherever possible. Source Data files will be made available to reviewers during evaluation of revised manuscripts and, if your paper is eventually published in JCB, the files will be directly linked to specific figures in the published article.

B. FINAL FILES:

****The license to publish form must be signed before your manuscript can be sent to production. A link to the electronic license to publish form will be sent to the corresponding author only. Please take a moment to check your funder requirements before choosing the appropriate license.****

Thank you for this interesting contribution, we look forward to publishing your paper in Journal of Cell Biology.

Sincerely,

Anna Huttenlocher, MD
Monitoring Editor
Journal of Cell Biology

Dan Simon, PhD
Scientific Editor
Journal of Cell Biology

Reviewer #1 (Comments to the Authors (Required)):

The revised version of the paper is much easier to follow at the level of the text and data presentation. The findings are important and of interest for readers of the Journal. I therefore believe that the manuscript should be published.

A few minor points the authors could consider modifying:

language-

Line 219 - ..."expressing a broad expression of..." - change

Line 362 - ..."One mechanistic basis..." -> something like "A possible mechanism that could account for these observations is..."

Line 364 - "A further mechanistic basis...." -> something like "An additional mechanism responsible for the effect of blebbistatin on the robustness of the polarity ..."

Data presentation-

I guess the authors have already considered it, but employing pseudocolors instead of the grey scale could have helped appreciating the results, especially in the movies.

Discussion-

It would be useful to include a few sentences directing the reader to related findings in other contexts. There are similarities and differences between the results reported on here regarding gradient interpretation, and mechanisms suggested for other cell types and migration modes. Recent papers I believe are worth mentioning are -

-Dynamic buffering of extracellular chemokine by a dedicated scavenger pathway enables robust adaptation during directed tissue migration. *Dev Cell*, 52 (2020), pp. 492-508 e10

-A negative-feedback loop maintains optimal chemokine concentrations for directional cell migration. *Nat Cell Biol*, 93 (2020), pp. 11-18

-Chemokine-biased robust self-organizing polarization of migrating cells in vivo. *Proc Natl Acad Sci USA*, 118 (2021)

Reviewer #2 (Comments to the Authors (Required)):

The authors have generally addressed my comments, but I have a few remaining concerns.

1. Because the authors have now removed their individual cell traces and only give the aggregate metrics of parameters like actin polarity in the main figures, I'm worried that readers will be confused about the images and statements of rear lifeact enrichment (which should give values of polarity less than one) whereas the aggregate metrics are basically above 1 throughout. I know the single cell traces are in the supplement, but it would be helpful to at least give the polarity values on all the representative individual cell images in the main figures.
2. So the casual reader does not equate rear enrichment of lifeact as an actual rear enrichment of actin polymer but rather just a surrogate for flow, the figures should say lifeact polarity throughout. I suspect a phalloidin-stained cell would still show leading edge enrichment of actin, so important that readers don't confound lifeact with the actual actin distribution, as most readers will.
3. Because the CK666 and blebbistatin drugs are used at sub-maximal concentrations, I'd like to see a caveat added that relative differences (or absences) of phenotype could be due to sub-maximal inhibition of their targets. It would be helpful to explain how the sub-maximal doses were chosen. And it would be useful to thread in the discussion of what it means that Arp2/3 is required for overall movement but intermediate concentrations give partial defects in gradient sensing.
4. While it would, of course, been nicer to measure actin flows directly rather than use the rear-enrichment of lifeact and utrophin, I understand the technical limitations. Still, it would be worth including the caveat that some of the shifts in distribution of these probes could reflect the organization of actin rather than the flows, since it is clear that the specificity of these various actin probes is quite complex and still somewhat poorly defined.
5. Given how indirect many of the perturbations in the paper are (partial inhibition of Arp2/3 and myosin, leading to partial effects on protrusion and contractility), several points of the discussion and abstract are overstatements. "inhibition of leading edge expansion" should be "partial inhibition", "inhibition of contraction" should be "partial inhibition of contraction". I'd like to see "suggest" rather than "indicate" given how indirect these perturbations are. Similarly, second to last sentence of abstract should be "suggesting that pure spatial sensing..." The last sentence of abstract should be "our data suggest" rather than "we show". Same points for end of the intro.

Reviewer #3 (Comments to the Authors (Required)):

The manuscript is improved and the authors either provide an explanation and/or changes to the manuscript for most issues. However, we still have a few concerns with the responses for points 7, 10, 13 and 16. In regards to points 13 and 16, no additional experiments were done to measure actin flow because of technical challenges and because other markers did not show actin flows. However, this seems to be an important component of the paper. The authors state that LifeAct enrichment correlates with actin flow but this is shown indirectly. There is no imaging of myosin and LifeAct together in the manuscript. Since they rely on only the one myosin assay for actin flows, which is not done the same way as their other laser wounding experiments, it would seem that the authors should either not include this feature as a major component of the model or should provide additional support for it. For example, testing how CK666 and blebbistatin affect actin flows would clarify the role of actin flows and imaging of myosin with other markers would clarify this as well.

Individual points:

1. Addressed
2. There is more explanation for the trends between the different conditions, but it seems that the trend for the cosine of theta beyond 100 microns actually increases at the cells get further away in both Pre-LW and Post-LW- are the authors implying that these cells are directing due to tissue geometry as well in this case or that they are not in the fin? The response was a bit confusing.
3. The text has clarified this and allows the reader to more easily interpret data - I would suggest adding additional labels for delta graphs in 1G to include persistent steps small turns, U-turns since this will be difficult for the readers to interpret without the text right by the side.
4. Addressed
5. Addressed
6. Addressed
7. While they say that they don't mention the phases until Figure 3 in the rebuttal, the phases are still mentioned in Figure 2- so the authors did not address this issue. They also did not mention how consistent the speeds are that characterize these phases. The two phases make it sound like this is time-dependent relative to the laser wound later in the paper, but the analysis in figure 2, but (to my understanding) they are looking at the timelag between different parameters and not assessing timing relative to the laser wound. Since I thought that they define the slow and fast phases as relative to the laser wound, it is confusing how they define the phases in Figure 2. Might they replace this with slower and faster speeds to more accurately describe the results here?
8. Addressed
9. Is there a threshold for the onset of movement in terms of microns displaced?
10. Addressed but it seems that there is no difference between pre and post LW in terms of surface area? The authors should

address this since this doesn't fit with their model where they mention there is growth of the leading edge in the search phase.

11. This clarifies this analysis

12. Addresses that it is a technical issue/sounds like they cannot do timing relative to mechanical wounding/amputation.

13. It would be beneficial to include why other markers do not show these actin flows in the manuscript/how we can interpret these results

14. Addressed

15. Addressed

16. Addressed for protrusions, not for actin flow

17. One possible explanation provided

Reviewer #1 (Comments to the Authors (Required)):

The revised version of the paper is much easier to follow at the level of the text and data presentation. The findings are important and of interest for readers of the Journal. I therefore believe that the manuscript should be published.

A few minor points the authors could consider modifying:

language-

Line 219 - ..."expressing a broad expression of..." - change

Line 362 - ..."One mechanistic basis..." -> something like "A possible mechanism that could account for these observations is..."

Line 364 - "A further mechanistic basis..." -> something like "An additional mechanism responsible for the effect of blebbistatin on the robustness of the polarity ..."

We have amended these expressions

Data presentation-

I guess the authors have already considered it, but employing pseudocolors instead of the grey scale could have helped appreciating the results, especially in the movies.

We indeed considered this but, since Lifeact is fused to a red fluorescent protein (Ruby), the contrast of red with black is not ideal for illustrating the differences and choosing another colour might have been confusing in terms of which label is used.

Discussion-

It would be useful to include a few sentences directing the reader to related findings in other contexts. There are similarities and differences between the results reported on here regarding gradient interpretation, and mechanisms suggested for other cell types and migration modes. Recent papers I believe are worth mentioning are -

-Dynamic buffering of extracellular chemokine by a dedicated scavenger pathway enables robust adaptation during directed tissue migration. Dev Cell, 52 (2020), pp. 492-508 e10

-A negative-feedback loop maintains optimal chemokine concentrations for

directional cell migration. *Nat Cell Biol*, 93 (2020), pp. 11-18

-Chemokine-biased robust self-organizing polarization of migrating cells in vivo. *Proc Natl Acad Sci USA*, 118 (2021)

We have included relevant comments in lines 491-494 and 505-509.

Reviewer #2 (Comments to the Authors (Required)):

The authors have generally addressed my comments, but I have a few remaining concerns.

1. Because the authors have now removed their individual cell traces and only give the aggregate metrics of parameters like actin polarity in the main figures, I'm worried that readers will be confused about the images and statements of rear lifeact enrichment (which should give values of polarity less than one) whereas the aggregate metrics are basically above 1 throughout. I know the single cell traces are in the supplement, but it would be helpful to at least give the polarity values on all the representative individual cell images in the main figures.

We added Lifeact polarity values in Figure 2 C and Figure S2 C.

2. So the casual reader does not equate rear enrichment of lifeact as an actual rear enrichment of actin polymer but rather just a surrogate for flow, the figures should say lifeact polarity throughout. I suspect a phalloidin-stained cell would still show leading edge enrichment of actin, so important that readers don't confound lifeact with the actual actin distribution, as most readers will.

We replaced word 'actin polarity' with 'Lifeact polarity' or 'Utrophin polarity' in all figures and in the text.

3. Because the CK666 and blebbistatin drugs are used at sub-maximal concentrations, I'd like to see a caveat added that relative differences (or absences) of phenotype could be due to sub-maximal inhibition of their targets. It would be helpful to explain how the sub-maximal doses were chosen. And it would be useful to thread in the discussion of what it means that Arp2/3 is required for overall movement but intermediate concentrations give partial defects in gradient sensing.

We included relevant comments in lines 354-358 and 470-473

4. While it would, of course, been nicer to measure actin flows directly rather than use the rear-enrichment of lifeact and utrophin, I understand the technical limitations. Still, it would be worth including the caveat that some of the shifts in distribution of these probes could reflect the organization of actin rather than the flows, since it is clear that the specificity of these various actin probes is quite complex and still somewhat poorly defined.

We added this in lines 243-245.

5. Given how indirect many of the perturbations in the paper are (partial inhibition of Arp2/3 and myosin, leading to partial effects on protrusion and contractility), several points of the discussion and abstract are overstatements. "inhibition of leading edge expansion" should be "partial inhibition", "inhibition of contraction" should be "partial inhibition of contraction". I'd like to see "suggest" rather than "indicate" given how indirect these perturbations are. Similarly, second to last sentence of abstract should be "suggesting that pure spatial sensing..." The last sentence of abstract should be "our data suggest" rather than "we show". Same points for end of the intro.

We added relevant modifications in the abstract, end of introduction and discussion.

Reviewer #3 (Comments to the Authors (Required)):

The manuscript is improved and the authors either provide an explanation and/or changes to the manuscript for most issues. However, we still have a few concerns with the responses for points 7, 10, 13 and 16. In regards to points 13 and 16, no additional experiments were done to measure actin flow because of technical challenges and because other markers did not show actin flows. However, this seems to be an important component of the paper. The authors state that LifeAct enrichment correlates with actin flow but this is shown indirectly. There is no imaging of myosin and LifeAct together in the manuscript. Since they rely on only the one myosin assay for actin flows, which is not done the same way as their other laser wounding experiments, it would seem that the authors should either not include this feature as a major component of the model or should provide additional support for it. For

example, testing how CK666 and blebbistatin affect actin flows would clarify the role of actin flows and imaging of myosin with other markers would clarify this as well.

To address this comment we included data we have available with Lifeact label in neutrophils with same mechanical wound assay, which show the same pattern of correlation between Lifeact distribution and speed (Figure S2, C and D). This links the rear Lifeact distribution with fast speed and fast flows in the same migration assay.

Individual points:

2. There is more explanation for the trends between the different conditions, but it seems that the trend for the cosine of theta beyond 100 microns actually increases at the cells get further away in both Pre-LW and Post-LW- are the authors implying that these cells are directing due to tissue geometry as well in this case or that they are not in the fin? The response was a bit confusing.

We mean that they are not in the fin so they have different tissue geometry biases. For example, at distances lower than 100 μ m cells are mostly within the fin and in this tissue there seems to be bias towards the wound as you get closer to the wound. This could be due to fin geometry or the LTB₄ initial gradient. Beyond 100 μ m cells are more likely in the CHT, outside the fin, and they could have other chemical or physical biases within that tissue that happen to be increasing as neutrophils move further from the wound. We put a new comment in manuscript in lines 153-155 to make this clear.

3. The text has clarified this and allows the reader to more easily interpret data - I would suggest adding additional labels for delta graphs in 1G to include persistent steps small turns, U-turns since this will be difficult for the readers to interpret without the text right by the side.

We have done this now, both in Figure 1 G and Figure S1 F.

7. While they say that they don't mention the phases until Figure 3 in the rebuttal, the phases are still mentioned in Figure 2- so the authors did not address this issue. They also did not mention how consistent the speeds are that characterize these phases. The two phases make it sound like this is time-dependent relative to the laser wound later in the paper, but the analysis in figure 2, but (to my understanding) they are looking at the timelag between

different parameters and not assessing timing relative to the laser wound. Since I thought that they define the slow and fast phases as relative to the laser wound, it is confusing how they define the phases in Figure 2. Might they replace this with slower and faster speeds to more accurately describe the results here?

The phases are only defined in relation to the gradient in Figure 3. In Figure 2 we are describing different states of cells, associated to slow and fast motion respectively, but these are not linked to the timing of gradient exposure and thus not defined as stages of gradient sensing. We revised the text to make this clearer using different wording than 'phase' prior to Figure 3.

9. Is there a threshold for the onset of movement in terms of microns displaced?

We included in the methods the threshold used for determining the beginning of motion (lines 675-680).

10. Addressed but it seems that there is no difference between pre and post LW in terms of surface area? The authors should address this since this doesn't fit with their model where they mention there is growth of the leading edge in the search phase.

We do not necessarily expect a difference in growth of leading edge pre- and post-wound because we envisage that the cells will undergo search and run phases in a non-synchronised manner when they are un-specifically migrating in the tissue. We reason that cells may have similar slow/searching phases both pre- and post-wound but they are only productive in oriented turns and directional runs in the presence of wound. We put a new comment in manuscript in lines 407-411.

13. It would be beneficial to include why other markers do not show these actin flows in the manuscript/how we can interpret these results.

We included a relevant comment in lines 222-227.

16. Addressed for protrusions, not for actin flow

Indeed we were not able to address this directly due to technical limitations, but we provide indirect evidence suggesting that the directional bias on cell

speed (which generally correlates with speed of retrograde flows) is affected by Blebbistatin and not CK666 (Figure 5 F).